# Unifying Specialized Visual Encoders for Video Language Models

Jihoon Chung [* 1]   Tyler Zhu [* 1]   Max Gonzalez Saez-Diez [1]
Juan Carlos Niebles [2]   Honglu Zhou [2]   Olga Russakovsky [1]

## Abstract

Recent advances in vision backbones have yielded powerful and diverse visual and video encoders. Yet, current Video Large Language Models encode visual inputs using an encoder from a single backbone family, limiting the amount and type of visual information they can process. We propose MERV, a Multi-Encoder Video Representation, which utilizes multiple encoders for a comprehensive video representation. To optimize heterogeneous features from a broad spectrum of encoders and ensure efficient and coherent feature integration, MERV first aligns encoder features spatio-temporally, then projects them into a unified structure, and finally fuses them through cross-attention. Under fair comparison, MERV achieves up to 4.62% higher accuracy than its base model, while introducing minimal extra parameters and training faster than equivalent single-encoder methods after parallelizing visual processing. Qualitative analysis shows MERV successfully captures and integrates domain knowledge from each encoder, opening new possibilities for scaling enhanced video understanding.

## 1. Introduction

Inspired by the sophisticated reasoning abilities of recent Large Language Models (LLMs) (Chiang et al., 2023; Chowdhery et al., 2023; OpenAI, 2023), researchers have focused on using them in many other domains to great success. The video counterparts, known as Video Large Language Models (VideoLLMs) (Bain et al., 2021; Li et al., 2023c; Lin et al., 2024; Luo et al., 2023; Maaz et al., 2024; Yu et al., 2024b), connect pre-trained vision encoders to LLMs by training a modality bridge from the vision space to the

*Equal contribution [1]Department of Computer Science, Princeton University, Princeton, NJ, United States [2]Salesforce Research, Palo Alto, CA, United States. Correspondence to: Jihoon Chung, Tyler Zhu <{jc5933,tylerzhu}@princeton.edu>.

*Proceedings of the 42nd International Conference on Machine Learning*, Vancouver, Canada. PMLR 267, 2025. Copyright 2025 by the author(s).

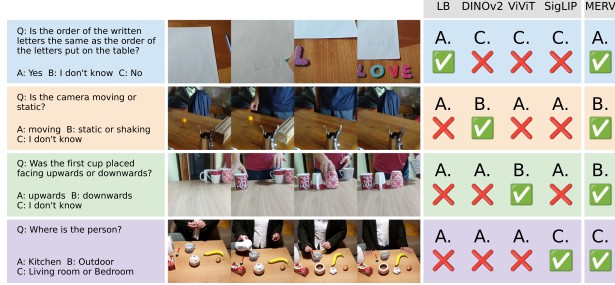

*Figure 1.* Examples where a single encoder model is the only model to correctly answer Perception Test questions (Pătrăucean et al., 2023), while MERV can correctly answer all types.

language space, allowing for reasoning to happen in the highly expressive language domain.

Most multimodal LLMs, such as LLaVA (Liu et al., 2023) for images and Video-LLaVA (Lin et al., 2024) for videos, opt for contrastively pre-trained encoders like CLIP (Radford et al., 2021) and LanguageBind (Zhu et al., 2024a). Their vision-language pre-training naturally lends itself as a bridge between the vision input and the LLM, circumventing the need to train heavy vision-language alignment modules like a QFormer (Li et al., 2022). These encoders are almost always pre-trained separately and vary in architecture, training data, and optimization strategy. Consequently, the features extracted by these encoders exhibit unique characteristics, each with inherent strengths and limitations. Contrastive encoders like CLIP may be better suited with their multimodal semantic alignment, but are inferior to models such as DINOv2 (Oquab et al., 2023) at fine-grained object level understanding. They also fail to take advantage of models trained specifically on videos, such as ViViT (Arnab et al., 2021). Despite this clear tension between vision backbones, previous research in VideoLLMs has relied on *only* one vision encoder for visual processing as one was thought to be sufficient for visual understanding, and already difficult enough to achieve vision-language alignment with. Any more encoders was unnecessary and not an effective tradeoff of runtime for compute.

In this paper, we argue that this choice to not use multiple encoders in existing VideoLLMs unnecessarily restricts their capabilities. For example, in Figure 1 we can see

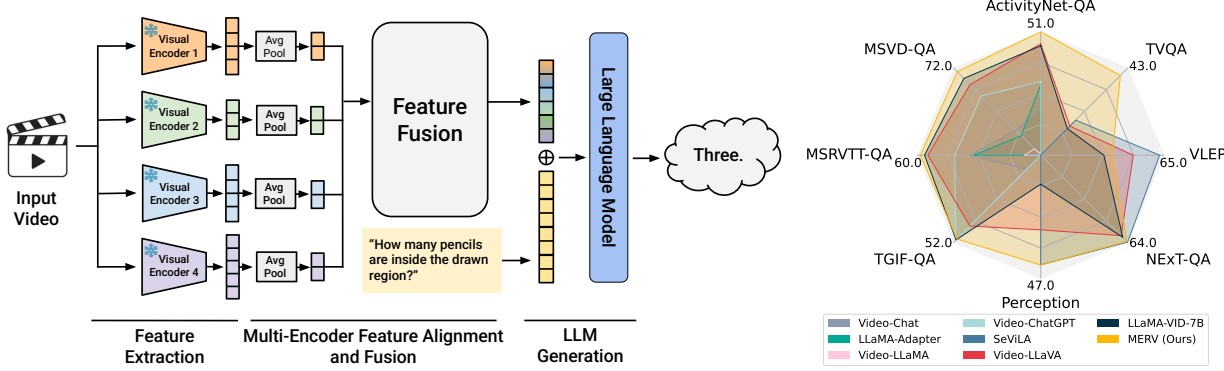

*Figure 2.* **MERV architecture and performance.** (**Left**) MERV proceeds in three main stages. First, we feed in our input video into each of visual encoders to get different representations. They are then spatio-temporally aligned before being fused by a cross-attentive mixer. The output is a visual embedding with an additive mix of information from all the encoders, which is combined with the text query to generate the result. (**Right**) We only compare MERV to the 7B prior models trained with comparable training data mixes. MERV is on-par or better than these single-encoder prior works, while deliver considerable gains over Video-LLaVA (Lin et al., 2024), from which MERV was initially adapted to form a Multi-Encoder Video Representation.

cases where only one of four different single-encoder models answers a given question correctly. While simple scene descriptions can be answered by image-level models, other questions require temporal and action-level comprehension, benefiting from features encoded with video models like ViViT (Arnab et al., 2021). Consequently, the reasoning capabilities of these VideoLLMs are directly limited by the inherent weaknesses of their respective pre-trained encoders. Therefore, employing multiple encoders could allow us to complement one encoder's weaknesses with another encoder's strengths. It also offers a cheap way to broaden the input distribution, leveraging different encoder's training mixtures. The wide adoption of the LLaVA paradigm is also indication that vision-language alignment is simple to achieve, even without language-aware vision models.

We propose *MERV*, a Multi-Encoder Representation of Videos, as a new method for integrating multiple visual encoders into a single VideoLLM using a cross-attentive encoder mixer for fusing representations. We introduce a spatio-temporally aligned representation for mixing the information from multiple types of visual encoders. Given the computational complexity of video tasks, we carefully experiment with optimization strategies and parallelizing the visual experts, allowing us to combine four distinct visual encoders with minimal computational overhead. Our frozen method outperforms all individual encoder methods, is up to 4.6% better than prior works (Lin et al., 2024) on video reasoning benchmarks, i.e., from 37.66% to 42.28% on TVQA (Lei et al., 2018), and on par with SeViLA (Yu et al., 2024b) on Perception Test (Pătrăucean et al., 2023), a challenging perception and reasoning diagnostic for video models. Finetuning the full model improves MERV past SeViLA (Yu et al., 2024b) by 2.2%, from 46.2% to 48.4%. Finally, we do a detailed qualitative study of our model's

capabilities on the Something-Something v2 dataset (Goyal et al., 2017). We show that MERV can accurately capture both contrastive encoders' (Zhai et al., 2023; Zhu et al., 2024a) strengths on general vision-language understanding, as well as ViViT's (Arnab et al., 2021) specialty on temporally-sensitive tasks (e.g. distinguishing pushing left vs. right), without trading off performance between these specializations as single encoder models do. [1]

## 2. Related Works

**VideoLLMs** build upon the powerful reasoning capabilities of LLMs by utilizing them as language decoders to enable instruction-followed video understanding. Key advancements include VideoChat (Li et al., 2023c) and Video-LLaMA (Zhang et al., 2023) for chat-based video understanding, LLaMA-Adapter (Zhang et al., 2024b) for pre-alignment, Valley (Luo et al., 2023) with multilingual LLMs, InternVideo (Wang et al., 2022) with a dedicated video encoder training phase, and Video-ChatGPT (Maaz et al., 2024) combining video-adapted encoders with LLMs. GPT4Video (Wang et al., 2023) supports video understanding and generation, while MovieChat (Song et al., 2024) focuses on long video comprehension. Models like Chat-UniVi (Jin et al., 2024), LLaMA-VID (Li et al., 2024b), and VTM (Lee et al., 2024) optimize token usage for video representation. Other notable models include Vamos (Wang et al., 2024b), which flexibly uses visual embeddings, action labels, and video captions as input; VideoChat2 (Li et al., 2024a), developed through three-stage progressive training; Video-LLaVA (Lin et al., 2024), which aligns image and video representations before projecting them to the

---

[1] Our code and pretrained weights are available at https://github.com/princetonvisualai/merv.

LLM space; and VideoPrism (Zhao et al., 2024), which also further trains a video encoder through masked distillation. Specialized models like VTimeLLM (Huang et al., 2024) focus on fine-grained video moment understanding and time-bound reasoning, while models like Elysium (Wang et al., 2024a) and Merlin (Yu et al., 2024a) can predict object trajectories. SeViLA (Yu et al., 2024b) uses LLM for frame localizer of the video for multiple-choice tasks. Finally, recently LLaVA-Hound-DPO (Zhang et al., 2024a) explored using DPO and a higher quality training set for better instruction following. Distinct from these aforementioned works, our approach centers on utilizing a diverse array of visual and video encoders, which are RGB-based but coming from different visual backbone families, each with its own unique strengths, to significantly enhance the capabilities of the VideoLLM framework. By strategically using these specialized encoders, we aim to capture a broader spectrum of visual information, enriching VideoLLMs' understanding of videos.

**Combining multiple encoders for multimodal LLMs** is gaining attention. Eyes Wide Shut (Tong et al., 2024b) explored mixing DINOv2 and CLIP features for LLaVA, but their results signal that mixing features effectively requires investigation. Both Mipha (Zhu et al., 2024b) and Prismatic-VLMs (Karamcheti et al., 2024) found that image encoders like CLIP and SigLIP, which are trained using vision-language contrastive loss, surpass other image encoders such as ViT and DINOv2, with SigLIP showing further improvements over CLIP. SPHINX-X (Gao et al., 2024) and SPHINX (Lin et al., 2023) combines multiple image encoders by concatenating features along the channel dimension, while BRAVE (Kar et al., 2024) concatenates features from multiple encoders sequence-wise, followed by a QFormer with masked modeling. More recently, Cambrian-1 (Tong et al., 2024a) creates a multimodal LLM which spatially aligns their inputs across different resolutions. This also reproduces our finding of the importance of spatio-temporal alignment. There is also the popular body of research on multimodal LLMs using many modalities including image, video, audio and/or 3D (Chen et al., 2023; Han et al., 2023; Li et al., 2023a; Lyu et al., 2023; Panagopoulou et al., 2023; Su et al., 2023; Sun et al., 2023a; Zhang et al., 2023; Liu et al., 2024a; Han et al., 2024; Liu et al., 2024b; Jain et al., 2024). In contrast, this paper dives into the video-language domain, exploring combining multiple image and video encoders and exploiting their structural similarities. Our multi-encoder feature alignment and fusion are both performant and efficient in FLOPs, and results in an all-encompassing additive mixture of features which previous works could not achieve.

## 3. MERV: Multi-Encoder Representation of Videos

Our goal for MERV is to systematically build a video model that leverages multiple encoders with an LLM to process a video following the LLaVA/PrefixLM (Liu et al., 2023; 2018) paradigm (see Figure 2). Unlike previous works, our focus is not on combining multiple encoders of different *modalities* that make use of additional information (depth, audio, etc) (Bachmann et al., 2022; Zhu et al., 2024a), but instead focus on RGB-encoders trained on different datasets and objectives that offer *different* visual understanding. We extensively ablate three key aspects to make this possible: our selection of *multiple* encoders, i.e., which visual encoders and how many to use (Sec 3.1); how we *align* the spatio-temporal representations of each encoder to mix the information together, especially in an efficient manner (Sec 3.2); and our *implementation* efficiencies, from the parallel visual processing to the training recipes (Sec 3.3).

### 3.1. Multi-Encoder Feature Extraction

Our final architecture uses four distinct types of models: spatial experts, fine-grained temporal experts, image-language experts, and video-language experts. We found experimentally that our choice of four performed the best across all types of questions, and ablate our choices in Sec. 4.2. More details about these four encoders are in Appendix Table 4.

**Spatial expert:** DINOv2 (Oquab et al., 2023) is trained using unsupervised learning on local-to-global correspondences in image data. The resulting features have robust object part understanding, as well as semantic image understanding, but can suffer from poor language grounding.

**Temporal expert:** ViViT (Arnab et al., 2021) is trained using supervised learning on short videos. The architecture is designed for modeling the interactions between frames using spatial and temporal attention, which lets it capture longer temporal dependencies than pure image models can.

**Image-Language contrastive expert:** SigLIP (Zhai et al., 2023) is trained using sigmoid contrastive learning on image-text pairs. The model is designed to learn a joint embedding space for images and text, which makes it good at understanding vision-language associations. However, it can overlook the finer details of an image which are not well described by text in its training data.

**Video-Language contrastive expert:** Finally, our video-language expert is LanguageBind (Zhu et al., 2024a). Used by Video-LLaVA (Lin et al., 2024), LanguageBind is trained through joint multimodal learning between text and multiple modalities (videos, infrared, etc.,) and understands the relationship between video and text and their high-level semantics. We only use the video encoder of LanguageBind.

## 3.2. Spatio-Temporally Aligned Representations for Feature Fusion

Our input is a batch of text, image-text, or video-text queries. The visual part of the input, either images or videos, is passed through each of the visual encoders to extract the respective features. Here we describe the detailed care we took in pre-processing to prepare the features for alignment.

First, images are treated as videos with repeated frames, so assume all inputs are videos from here on out. A video is of shape $T \times H \times W$, where $T$ is the number of frames and $H, W$ are the height and width of the frames, and produce an output of shape $t_e \times h_e \times w_e$ for an encoder $e$. One obstacle with using different visual encoders is that each model outputs features with a different structure. For example, given an input of shape $16 \times 224 \times 224$, ViViT outputs a feature of shape $8 \times 14 \times 14$ whereas LanguageBind's features are of shape $16 \times 16 \times 16$. Image-based encoders will not change the temporal dimension, whereas ViViT downsamples the frames by a factor of 2.

For temporal alignment, as each encoder is flexible enough to handle varying input frames, we simply choose our input $T$ for each encoder so that each output $t_e$ is the same across all encoders, i.e. $t$.

**Pre-fusion projection.** Now we need to achieve spatial alignment among the features. Naïvely combining them will not work as they all have different spatial shapes. Using the full resolution features would also be prohibitively expensive. We design a pre-fusion projector to tackle both issues by aligning and compressing the features.

Suppose our feature from encoder $e$ is $\mathbf{v}_e \in \mathbb{R}^{t \times h_e \times w_e \times d_e}$, where $d_e$ is the dimension of encoder $e$, and assume the output spatial representations are square (i.e. $h_e = w_e$, but we keep notation for clarity). Our pre-fusion projector uses an adaptive 2D average pool $\mathcal{P}$ for each encoder to resize the spatial dimensions to the same $h \times w$ for all encoders, where $h < h_e$ and $w < w_e$. As $t$ is the same across each $\mathbf{v}_e$, this *spatio-temporally aligns* the representations.

Finally, we need to connect the varying embedding dimensions $d_e$ to a same dimensional space. We add a linear layer to project the features from dimension $d_e$ to $d$, the LLM's dimension. In total, our pre-fusion projection is

$$\mathbf{x}_e := \mathcal{P}(\mathbf{v}_e)W_e \in \mathbb{R}^{\ell \times d} \quad \text{for } e \in \text{Encoders} \quad (1)$$

where $W_e \in \mathbb{R}^{d_e \times d}$ is each encoder's output linear layer, and $\ell = t \times h \times w$. This projector is lightweight, having only $d \times \sum_e d_e$ trainable parameters for dimension matching, making it easy to scale to an arbitrary number of visual encoders. For detailed ablations, see Section 4.2.1.

**Feature fusion strategies.** The final part of our pipeline is fusing the multi-encoder information together using cross-

attention with learnable queries to additively mix the different representations together. The visual features determine the weights of the linear mixture, which we find sufficient for our task. We use a single randomly initialized query $\mathbf{Q} \in \mathbb{R}^{1 \times d}$, keys as $\overline{\mathbf{X}} = [\overline{\mathbf{x}}_1 \ \dots \ \overline{\mathbf{x}}_N] \in \mathbb{R}^{N \times d}$, where $\overline{\mathbf{x}}_e \in \mathbb{R}^d$ is each encoder's features (after Eq. 1) averaged over the sequence dimension $\ell$ for a faster computation, and $N$ the number of encoders, and values as $\mathbf{X} = [\mathbf{x}_1 \ \dots \ \mathbf{x}_N] \in \mathbb{R}^{N \times \ell \times d}$. We calculate our final unified feature as

$$\mathbf{O} := \text{Softmax}\left(\frac{\mathbf{Q}\overline{\mathbf{X}}^\top}{\sqrt{d}}\right) \mathbf{X} \in \mathbb{R}^{\ell \times d}. \quad (2)$$

The final step is to concatenate the visual embedding and tokenized text together into the LLM. We use the base LLaMA-2 7B model (Touvron et al., 2023b), which we found performs better than the chat model. We test multiple alternate feature fusion strategies in Section 4.2.2.

### 3.3. Implementation Efficiencies

**Parallelized visual encoding.** At a first glance, using multiple encoders seems to be a large cost to pay when comparing the raw FLOPs and parameters. However, a key benefit of the LLaVA style architecture is that the entire feature extraction and projection pipeline can happen *in parallel*. To make this possible, we build on top of recent powerful advances in parallel processing for LLMs and use PyTorch's Fully Sharded Data Parallel (Zhao et al., 2023). As the video encoders themselves are much smaller than the LLM blocks, and their visual encoding processes are completed in roughly the same amount of time, most of the overhead in running four encoders is already covered by having just one encoder (*ref.* Figure 3). We provide some timing numbers in Section 4.2.3 and find that our step time is similar to that of the single-encoder methods.

Our code is built on top of the Prismatic VLM codebase (Karamcheti et al., 2024), which efficiently implements vision-language model (VLM) training. We add the ability to handle videos and an arbitrary number of visual encoders, along with many useful features for training. Our training is efficient for using multiple visual models, completing in under 24 hours using 8 L40-48GB GPUs, and down to 8 hours using 8 H100s. In contrast, the Video-LLaVA codebase runs the Stage-2 training in around 38 hours on the same L40 setup and could not easily support multiple encoders in our initial attempts.

**MERV frozen and full.** Many different recommendations for training LLaVA style models have been made since its inception. This is only made more complicated by the introduction of new datasets with every new VideoLLM architecture, making it difficult to properly determine the

*Table 1.* **Comparison of different multimodal LLMs on video reasoning benchmarks**. We employ ChatGPT to evaluate performance following Video-ChatGPT where applicable (version `gpt-3.5-turbo-0613`). * denotes our evaluation of using the author provided checkpoint. The first five datasets were used as evaluation test sets during development; the last three were held-out for our final evaluation.

| Methods | MSVD-QA | | MSRVTT-QA | | TGIF-QA | | Perception | ActivityNet-QA | | NExT-QA | VLEP | TVQA |
| --- | --- | --- | --- | --- | --- | --- | --- | --- | --- | --- | --- | --- |
| | Acc | Score | Acc | Score | Acc | Score | Acc | Acc | Score | Acc | Acc | Acc |
| *Alternative data mixes* | | | | | | | | | | - | | |
| Video-Chat (Li et al., 2023c) | 56.3 | 2.8 | 45.0 | 2.5 | - | - | - | 26.5 | 2.2 | - | - | - |
| LLaMA-Adapter (Zhang et al., 2024b) | 54.9 | 3.1 | 43.8 | 2.7 | - | - | - | 34.2 | 2.7 | - | - | - |
| Video-LLaMA (Zhang et al., 2023) | 51.6 | 2.5 | 29.6 | 1.8 | - | - | - | 12.4 | 1.1 | - | - | - |
| Video-ChatGPT (Maaz et al., 2024) | 64.9 | 3.3 | 49.3 | 2.8 | - | - | - | 35.2 | 2.7 | - | - | - |
| SeViLA (Yu et al., 2024b) | - | - | - | - | - | - | 46.2 | - | - | **63.6** | **64.4** | 38.2 |
| LLaMA-VID-7B* (Li et al., 2024b) | 69.30 | 3.74 | 57.84 | 3.24 | 51.31 | 3.26 | 41.64 | 46.45 | 3.22 | 60.61 | 57.65 | 37.43 |
| LLaMA-VID-13B* (Li et al., 2024b) | 70.25 | 3.77 | 58.58 | 3.26 | 51.26 | 3.26 | 41.54 | 46.79 | 3.23 | 60.03 | 61.98 | 41.33 |
| *Same data mixes* | | | | | | | | | | - | | |
| Video-LLaVA* (Lin et al., 2024) | 67.74 | 3.69 | 56.90 | 3.18 | 47.99 | 3.17 | 44.22 | 47.08 | 3.27 | 59.61 | 61.21 | 37.66 |
| MERV (frozen) / MERV | **70.97** | 3.76 | **59.03** | **3.25** | 51.1 | 3.26 | 46.21 | **50.87** | **3.34** | 63.09 | 58.66 | **42.28** |
| Gains to Video-LLaVA* | +3.23 | +.07 | +2.13 | +.07 | +3.11 | +.09 | +1.99 | +3.79 | +.07 | +3.48 | -2.55 | +4.62 |
| MERV (full) | 70.48 | **3.79** | 57.25 | 3.24 | 51.39 | 3.28 | **48.41** | 49.93 | 3.33 | 61.36 | 60.07 | 39.42 |
| Gains to Video-LLaVA* | +2.74 | +.10 | +0.35 | +.06 | +3.40 | +.11 | +4.19 | +2.85 | +.06 | +1.75 | -1.14 | +1.76 |

best recipe for one's own setup. We intentionally fix our dataset to be the same as Video-LLaVA's so we can isolate the impacts of the training setup, from which we find two viable settings: *MERV (frozen)* and *MERV (full)*. The original Video-LLaVA's recipe has two stages: a Stage 1 pre-training on captioning data to align only the connector between the pre-trained vision encoder and LLM, and a Stage 2 instruction tuning on both connector and the LLM.

- *MERV (frozen)*, which performs only the Stage 2 instruction tuning inspired by Karamcheti et al. (2024); it achieves similar results to the original Video-LLaVA recipe in only 43% of the time;
- *MERV (full)*, which undergoes both of the Stage 1 captioning pre-training and the Stage 2 instruction tuning, but also unfreezes the LLM during Stage 1 for a slight improvement on a few benchmarks.

As MERV (frozen) is faster to train with similar performance, we adopt that recipe by default for analysis, and interchangeably use *MERV* to refer to it for simplicity from here on out. Detailed analysis is provided in Section 4.2.3.

## 4. Experimental Results

In this section, we show that our method outperforms prior works across standard video-language benchmarks before moving onto an in-depth analysis of our method and its specializations in the next section.

**Datasets and training procedure details.** For fair comparison, our data mix is the same as Video-LLaVA (Lin et al., 2024). The Stage 1 data is single-turn concise captioning, with 558k (image, text) pairs from LAION filtered by LLaVA (Liu et al., 2023) and 702k (video, text) pairs from Valley (Luo et al., 2023). The Stage 2 data is multi-turn conversations, detailed captioning and reasoning, with 665k (image, text) pairs from LLaVA and 100k (video, text)

instructions from Video-ChatGPT (Maaz et al., 2024).

All the preprocessing, including frame extraction, adheres to the original method that each encoder is trained with. We extract 16 uniformly sampled frames from each video, except for ViViT which extracts 32 frames by default but produces a 16-frame output feature.

For MERV (frozen), we train on only Stage 2 data for 1 epoch with a learning rate of $2 \times 10^{-5}$ and a batch size of 128 with gradient accumulation. For MERV (full), we first train on Stage 1 data with a learning rate of $1 \times 10^{-4}$ and the projectors, feature fusion, and LLM unfrozen with similar settings. Both recipes use an initial warmup ratio of 0.03 and a cosine schedule.

**Evaluation.** We evaluate our model on a comprehensive suite of video understanding benchmarks, including the open-ended MSVD-QA (Xu et al., 2017), MSRVTT-QA (Xu et al., 2017), TGIF (Jang et al., 2017), and ActivityNet-QA (Yu et al., 2019), as well as the multiple-choice benchmarks NExT-QA (Xiao et al., 2021), VLEP (Lei et al., 2020), TVQA (Lei et al., 2018), and Perception Test (Pătrăucean et al., 2023). We emphasize that NExT-QA, VLEP, and TVQA datasets are **held-out** datasets that we did not use during our experiments, and only evaluated once after all the design is completed. We report both accuracy and score following the Video-ChatGPT evaluation protocol where applicable, and all evaluations are done zero-shot without any dataset-specific fine-tuning. Results using GPT-3.5-turbo for evaluation are done with the June 13th, 2023 cutoff date.

### 4.1. Comparison to State of the Art

Table 1 tabulates the performance of MERV (frozen) and (full). We compare our model to the existing works, including Video-LLaVA (Lin et al., 2024) that share our train-

*Table 2.* **Ablating design choices.** We highlight our defaults in orange and **bold** the best results. Average accuracy is on MSVD, MSRVTT, TGIF, and Perception Test. Full metric results are in the Appendix.

(a) **Pre-fusion projectors.** * is 16 frames instead of 8. Top two rows are projector-free baselines.

| Projector | Avg Acc | Params | FLOPs |
|---|---|---|---|
| 257 tok | 54.76 | - | - |
| class tok | 52.05 | - | - |
| 2D Avg | 54.96 | 0 | 2.1M |
| 2D Avg* | **55.86** | 0 | 4.2M |
| 2D Attn | 52.12 | 12.7M | 9.7G |
| 2D Conv | 54.23 | 237M | 241G |
| 3D Avg* | 55.09 | 0 | 4.2M |
| 3D Conv | 55.42 | 113M | 232G |

(b) **Pre-fusion output token.** We ablate the optimal token size per frame for the pre-fusion projector.

| Tkns | MSVD | MSRVTT | TGIF |
|---|---|---|---|
| 1 | 61.94 | 54.64 | 41.41 |
| 4 | 64.47 | 55.72 | 45.32 |
| 16 | 67.23 | 56.44 | 47.75 |
| 64 | **69.08** | **58.00** | **50.01** |
| 100 | 68.38 | 57.47 | 48.78 |
| 144 | 68.65 | 57.73 | 48.81 |
| 256 | 68.46 | 57.72 | 48.66 |

(c) **Feature fusion strategy.** Cross-Attn additive mixing is the best overall among all the strategies on accuracy, for its FLOPs.

| Strategy | Avg Acc | FLOPs |
|---|---|---|
| Cross-Attn | **56.83** | 17.19 T |
| Concat (Seq.) | 54.45 | 43.09 T |
| Concat (Ch.) | 56.64 | 16.29 T |
| Learnable W | 55.01 | 16.24 T |
| 25% - Mixed | 54.19 | 16.39 T |

ing data mixture, and other VideoLLMs (Li et al., 2023c; 2024b; Maaz et al., 2024; Yu et al., 2024b; Zhang et al., 2023; 2024b). We find that our method, generating video representations using multiple visual encoders that specialize in different skills of video understanding, outperforms Video-LLaVA across nearly all of the benchmarks, with a 4.1% gain on Perception Test, a 3.7% gain on ActivityNet and a 4.6% gain on TVQA. Both of our methods perform better overall than Video-LLaVA, even when using less data with just Stage 2 as shown by the MERV (frozen) numbers. While MERV (full) is not a strict improvement to MERV, it still improves on some difficult benchmarks (e.g., Perception Test) with its additional video-language alignment. We believe that this makes MERV (full) generalize better in unseen settings outside of these testing benchmarks, and recommend using this recipe when possible. Compared to LLaMA-VID-7B, which uses a different training mix, we are better in nearly all benchmarks, up to around 4.5% across Perception Test, ActivityNet, and TVQA. Moreover, MERV (full) outperforms SeViLA on the Perception Test zero-shot with 48.4%, compared to 46.2%. Overall, our design shows a significant improvement over Video-LLaVA and prior methods as a whole.

## 4.2. Ablations

In this section, we justify our design choices for the projectors, feature fusion strategies, and training recipes. Our ablations are done with the MERV (frozen) recipe.

### 4.2.1. PRE-FUSION PROJECTORS

The first module we investigate is our projectors, which serve to connect each encoder from its pre-trained embedding space to a common embedding space. We test two types of projectors: image-level, which operate on frames independently, and video-level, which aggregate information across frames. Projector details are provided in Section A.3 in the Appendix. We report average performance across our development sets of MSVD, MSRVTT, TGIF, and Perception Test.

**Pre-fusion projector.** Table 2a tabulates our projector ablation on average accuracy, parameter count, and FLOPs, with the default settings of LanguageBind as the single vision encoder and an 8 frame 64 token projection output. We find that 2D average pooling is the best, with 55.86% average accuracy, even better than using no projector on the full 257 token representation (as in Video-LLaVA (Lin et al., 2024)), with 54.76% average accuracy. It also has no trainable parameters and the fewest FLOPs. The projection serves as a form of feature selection, allowing the LLM to efficiently reason only about the most relevant information.

One result worth noting is the poor performance (52.12%, 3.7% lower) of the attentive resampler, a popular projector choice. The attentive resampler is less responsive to the input spatio-temporal structure compared to other projectors, which leads it to being a weaker projector for us. However, it would likely be better with higher quality data from which the transformer can learn. Increasing the frame resolution from 8 to 16 was also a large improvement, showing that increasing temporal resolution is still important. This highlights the importance of aligning representations with their spatial and temporal structure, *especially* for video models, which extract many more frames of visual information.

**Projector token length.** Similarly, we ablate the output token length of the projector. Table 2b tabulates the performance of producing 1 to 256 tokens for a 2D average pooling projector with 16 frames as input. We see that the performance peaks at 64 tokens with 69.08% MSVD accuracy, with worse performance for longer token lengths (max. 68.65%). This balances the number of tokens used for condensing the visual embedding while also minimizing the extra processing needed by the LLM.

### 4.2.2. FEATURE FUSION STRATEGIES

Next, we test different strategies for fusing the information from all of the features, with detailed breakdowns in Table 2c. First, we evaluate two popular concatenation methods, in either the token sequence dimension, or the channel

dimension followed by an MLP for matching the LLM dimension. While sequence-wise concatenation is widely used in multimodal LLMs (Tong et al., 2024b), our method outperforms it while using significantly less computation, with a 56.8% average accuracy compared to 54.4%, while also using 2.5× fewer FLOPs. Concatenation channel-wise reaches a similar performance of 56.6% and a lightweight cost. However, our cross-attention shows better average performance, better or on par on 3 out of 4 benchmarks (*ref.* Appendix Table 6c), with the additional benefit of having accessible encoder weightings for analysis, so we stick with cross-attention for our final design. We also try different methods of additive mixing as an ablation. The last two rows of Table 2c show the performance when either learning the additive weights as a single scalar or by fixing the weights to be 0.25 for each of the 4 encoders. We see that using cross attention outperforms both methods by 1.8% and 2.6%, as our feature fusion module can dynamically generate better fused embeddings based on the visual input.

### 4.2.3. TRAINING RECIPES

Finally, we also compare different training recipes based on the literature and our own expertise. Apart from training recipes mentioned in Section 3.3, many recent works have attempted some combination of other strategies, such as unfreezing the vision encoders in Diao et al. (2024), which usually requires more training data.

We systematically map out this landscape, fixing our dataset to be the same as Video-LLaVA's. Unlike them, we found that the Stage 1 phase did not help much when training only the projectors and feature fusion (*ref.* Appendix Table 5), with roughly the same average accuracy. Stage 2 instruction tuning alone leads to similar results in 43% of the total time, so we adopt this recipe for efficiency and refer to it as *MERV (frozen)*. This recipe is still unsatisfying as it leaves a large amount of data, approximately 1.3M vision-text pairs, unused for training. In our empirical observations, we found that the resulting video-language alignment was suboptimal. The distributions of language used in video training datasets and benchmarks sparsely overlap based on their sentence embeddings, which could be impacting our ability to generalize zero-shot on downstream benchmarks. We address this by *unfreezing* the LLM during Stage 1 to better learn this alignment, improving performance on a few key benchmarks, especially Perception Test, by up to 2.2%. We call this recipe *MERV (full)*.

As another ablation, we train MERV on a single stage comprised of the Stage 1 and Stage 2 data mixed together (*ref.* bottom of Appendix Table 5). Surprisingly, this does worse than the explicit two stage training recipe. We attribute this to the explicit types of data in each stage being a form curriculum learning, showing that these stages are still im-

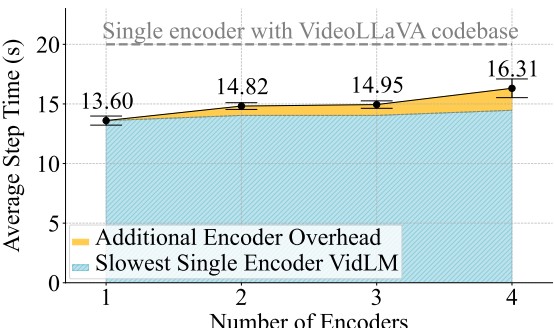

*Figure 3.* **Extra encoders incur minimal step time overhead**. We add encoders in the order of DINOv2, LanguageBind, SigLIP, ViViT, plotted alongside the slowest single encoder in each group.

portant for optimal performance.

Finally, we provide evidence for the efficiency of our method (*ref.* Figure 3). We use the default FSDP sharding strategy PyTorch provides; it is not currently possible to specify explicit plans for which modules go where (but may be possible as FSDP matures). However, even with this basic strategy, our method is dominated by the slowest single encoder present, incurring very little additional overhead from extra encoders due to this parallelization, making it cost-efficient to scale up in the number of encoders.

### 4.3. Ensemble Composition

The original motivation of our work was to choose encoders with complementary visual knowledge to form a comprehensive representation for our final model. The key questions are 1) do we benefit by using more than one encoder, and 2) do we need all four encoders, i.e. does each one meaningfully contribute to the final performance?

**Can we make use of more encoders?** The conventional wisdom is to use a single encoder, typically a contrastively trained vision-language model like CLIP, SigLIP, or LanguageBind (Radford et al., 2021; Zhai et al., 2023; Zhu et al., 2024a), in a VideoLLM. In Figure 4a (●), we show the four single encoder models corresponding to each of our chosen encoders using their full embeddings. They not only all perform worse than MERV but also use more FLOPs, as without our pre-fusion projectors, their sequence lengths are at least 4× ours.

**Are each of the encoders contributing?** To affirm that this set of four encoders is actually beneficial for improving understanding, we train three-encoder VideoLLMs under the same strategy, but removing a different encoder each time. Each of these models does worse based on the strength of the encoder removed, meaning that MERV is using their knowledge (*ref.* Fig. 4a, ▲). The minor drop in FLOPs illustrates how most of the computation is still dominated by the LLM, not the vision encoders.

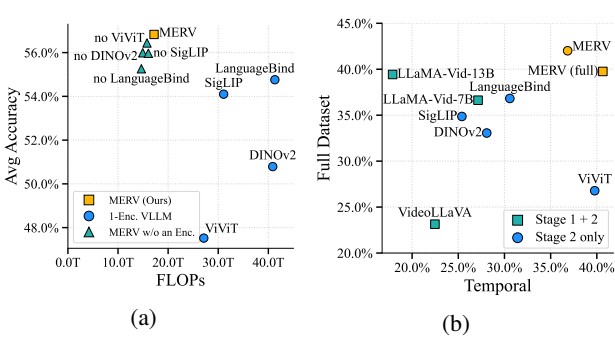

(a)           (b)

*Figure 4.* **Analysis plots supporting our design of multiple encoders, from their accuracy to their skill specializations.**
**(a) Visual Encoder Subsets.** MERV outperforms single-encoder VideoLLMs (●), with our feature projectors unlocking more computational efficiency. Removing any encoder also reduces MERV performance (▲). Average accuracy is across MSVD, MSRVTT, TGIF, and Perception Test. **(b) SSv2-MCQ and Temporal.** Temporal denotes performance on 12 selected classes where actions are indistinguishable if played in reverse (*ref.* Sec. 5.3). Full results are in the Appendix Tables 7, 8, 10.

# 5. Analysis

## 5.1. Cross Attention Weights Activate on Corresponding Videos

We first understand our method by looking at the cross attention weights on our 4 benchmark datasets (MSRVTT, TGIF, MSVD, and Perception Test), and visualizing the videos which have the highest attention weight for each of the encoders in Figure 5. This lets us see what types of videos activate each encoder the most. As expected, ViViT attention weights are highest on videos with large motion, as ViViT has strong temporal understanding. Meanwhile, SigLIP is utilized for videos that have textual data in the video, likely due to being vision-language contrastively trained, especially with textual data during training. DI-NOv2 and LanguageBind are both preferred by videos with static scenes, but LanguageBind is preferred for videos with foreground motion.

## 5.2. MERV Can Capture Visual Skills of Different Encoders

Next, we ask if our model effectively captures knowledge from its encoders. We first answer through our previous open-ended QA benchmarks. To assess the performance across different visual tasks, we create "pseudo"-skill categories by looking at the first word of the question sentence, which are often WH-words. They can be viewed as a proxy of skills required to solve the task. For example, `Where` requires spatial understanding and `When` requires temporal understanding. Figure 6 shows the relative performance of different visual encoders. While the contrastive models generally dominate each category, no single encoder performs

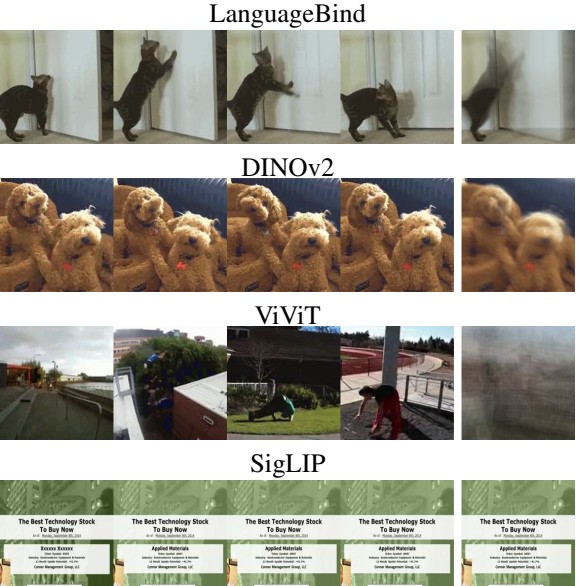

*Figure 5.* **Videos that give the highest attention weight for each of the encoders.** The right-most column shows the average frame of the video. For more examples, see Figure 11 in the Appendix.

best in all tasks. LanguageBind, for example, performs the best in TGIF-`What` with 46.23%, while DINOv2 performs on par with the best in MSVD-`Who` with 82.12%. Our method combines different encoders into an unified representation and consistently matches or improves the best-performing single-encoder model. Raw numbers are in Table 9 in the Appendix.

## 5.3. MERV Can Intuit Motion and General Understanding Simultaneously

We take another angle to quantifying how well our model learns from each of its individual encoders by looking back to classic video action recognition datasets. They are commonly used for general action understanding, but they can also be used to evaluate finer-grained capabilities. We are most interested in both general understanding and distinguishing actions which are temporally ambiguous, i.e., indistinguishable when reversed in time, such as *Pulling [something] from left to right* and *Pulling [something] from right to left*. This offers a fair analysis of both general and finer-grained video understanding.

We turn to the Something-Something v2 (Goyal et al., 2017) (SSv2) dataset, where the goal is to classify an input video into one of 174 classes, e.g., *Pulling [something] from left to right*. This allows us to analyze our model's understanding of temporal-spatial interaction with minimal distractions from scene understanding and real-world semantics. However, evaluating SSv2 as a zero-shot open-ended task is difficult with a long tail of specific categories. Thus we reformat the dataset into 5-way multiple-choice questions

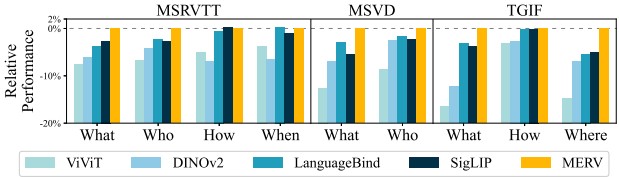

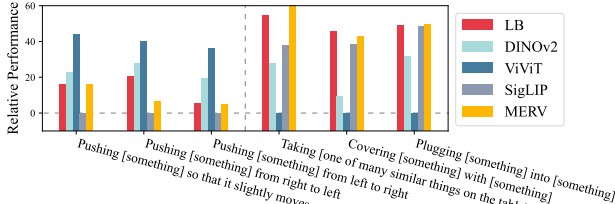

Figure 6. **Single encoder vs. MERV on different types of video tasks**. We plot the relative performance of VideoLLMs with different visual encoders. While each single encoder has its strength in different tasks, our method shows better performance than all the other single encoders in almost every task. We only plot tasks with more than 500 samples. See Appendix for details.

Figure 7. **Single-Encoder Performance Difference in Something-Something v2 - MCQ**. ViViT shows better performance on tasks where temporal understanding is crucial, while LanguageBind (LB) and SigLIP show better performance where task can be solved from single-frame understanding.

(MCQ) and fix the prompt to be "How is the object in the video being interacted with?". Incorrect choices were randomly sampled from the other 173 classes. We call this benchmark "SSv2 - MCQ" to distinguish it from the original classification task. Additionally, we selected 12 classes *a priori* from SSv2, where the action is indistinguishable if reversed in time, e.g., *Pulling [something] from left to right* and *Pulling [something] from right to left* to form a subset that can test the model's ability in temporal understanding.

Figure 4b plots the performance of MERV and single-encoder models on this temporal subset ($x$-axis) against the full dataset ($y$-axis). We see that the ViViT single-encoder model, which often falls short in other video QA benchmarks, surprisingly performs better than other encoders at 39.77% on temporal subset, which is 9.19% higher than the next closest model LanguageBind. However for the full SSv2 - MCQ, ViViT suffers with a worse performance of 26.78%, as ViViT's strength is on temporal understanding despite lacking in vision-language understanding. Contrastive encoders have the upper-hand in most other classes.

We believe that the architecture, datasets, and objective of each model causes these difference. ViViT processes spatial-temporal tubelets for embeddings, leading to better temporal understanding despite only being pre-trained on Kinetics-400 classification. SigLIP uses image-based ViT with no temporal layer has limited temporal understanding, but has a greater knowledge due to its larger training set and contrastive objective. MERV, at 42%, shows better performance compared to all these single-encoder models via leveraging strength of all the individual encoders. MERV (full) performs better than both VideoLLaVA (Lin et al., 2024) and the 7B and 13B variants of LLaMA-Vid (Li et al., 2024b).

Finally, we also plot the performances of six SSv2 classes where the performance difference between ViViT and SigLIP is largest in Figure 7. We see that actions which cannot be inferred from a single frame are the ones that ViViT performs better, e.g., *Pushing [something] from left to right* is indistinguishable from *Pushing [something] from right to left* if temporal information is omitted. Meanwhile, SigLIP performs better for classes where understanding the

semantics of the scene can hint the action that is happening, e.g., if a frame where a blanket is dropped on an object is given, one can easily expect *Covering [something] with [something]* without watching the full video. See Appendix Figure 9 for sample videos of the these classes.

## 6. Conclusion

Previous VideoLLMs have been limited to relying on a single visual model for feature extraction, which leads to limited understanding capabilities of vastly different video tasks. In our work, we break this paradigm and explore various fusion strategies for combining information from multiple visual experts to generate a representation that can leverage the capabilities of different video encoders. We find that our multi-encoder feature fusion is able to outperform comparable methods by up to 4.62% on video reasoning benchmarks. We show that the method can obtain better performance than the best-performing single-encoder model with minimal computational overhead. Finally, we quantitatively and qualitatively observe the skill specializations our model learns on an MCQ format of Something-Something v2, which confirms both that encoders can be specialized and that our model captures both axes of knowledge. Our paper proposes some initial steps in rethinking how we approach the use of multiple encoders. We hope that this inspires others to also consider this problem as another direction for scaling and improving their VideoLLMs.

## Acknowledgements

This work is supported by the National Science Foundation under Grant No. 2107048 and the Princeton First Year Fellowship to TZ. We also thank Allison Chen and William Yang for detailed comments and feedback.

## Impact Statement

Our work aims to facilitate video understanding, which can lead to positive social impacts such as a video captioning model for low-vision users, automatic detection of medical

emergencies, or better self-driving cars. It can also lead to negative social impacts like easy surveillance by the authorities, and human-like internet bots being used for scamming purposes. We follow the same safeguards implemented by the original authors of the datasets, the visual models, and the LLM models. We have not put any additional safeguards ourselves.

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

# A. Appendix / Supplemental Material

## A.1. Limitations

Our work is based on the LLM, LLaMA-2 7B model (Touvron et al., 2023b), and as with many other VideoLLM models, the performance of our method is hugely dependent on the capabilities of the LLM model, and better-performing models often demand significantly more computation. MERV requires an LLM and running multiple encoders, which can be computationally intensive and can lead to out-of-memory errors in resource-limited settings (while our efficient implementation alleviates the issue). While FSDP (Zhao et al., 2023) allows us to easily and effectively train larger models across multiple GPUs than would otherwise be possible, its generality also makes it difficult for us to design tailored sharding strategies that would maximize the performance of our model. However, with future improvements to data parallelism, our model can still benefit greatly and run even more efficiently. Also, despite the improved speeds, there is still an upper bound for what constitutes a *reasonable* training time that still allows us to test many of our design assumptions, which limits the scale and number of experiments we can run. While we show that our method can successfully leverage information from different visual encoders, nevertheless if the encoders themselves are limited in video understanding capability, MERV cannot fully compensate for that.

## A.2. Model Details

### A.2.1. BASELINE ENCODER AND LLM DETAILS

*Table 3.* **Visual Encoder and LLM Information**.

| Model | Visual Encoder | LLM |
|---|---|---|
| Video-Chat (Li et al., 2023c) | ViT-G (EVA-CLIP) (Sun et al., 2023b) | StableVicuna (contributors, 2023) |
| LLaMA-Adapter (Zhang et al., 2024b) | CLIP (Radford et al., 2021) | LLaMA-1 7B (Touvron et al., 2023a) |
| Video-LLaMA (Zhang et al., 2023) | ViT-G (EVA-CLIP) (Sun et al., 2023b) + BLIP-2 Q-Former (Li et al., 2023b) | Vicuna-7B v0 (Chiang et al., 2023) |
| Video-ChatGPT (Maaz et al., 2024) | CLIP (Radford et al., 2021) | Vicuna-7B v1.1 (Chiang et al., 2023) |
| SeViLA (Yu et al., 2024b) | ViT-G (EVA-CLIP) (Sun et al., 2023b) + BLIP-2 Q-Former (Li et al., 2023b) | FlanT5-XL (3B) (Chung et al., 2024) |
| LLaMA-VID-7B (Li et al., 2024b) | EVA-G (Fang et al., 2023) | Vicuna-7B v1.5 (Chiang et al., 2023) |
| LLaMA-VID-13B (Li et al., 2024b) | EVA-G (Fang et al., 2023) | Vicuna-13B v1.5 (Chiang et al., 2023) |
| Video-LLaVA* (Lin et al., 2024) | LanguageBind (Zhu et al., 2024a) | Vicuna-7B v1.5 (Chiang et al., 2023) |

### A.2.2. MERV ENCODER DETAILS

Here, we detail the visual encoder details, LLM, and the training objectives. **We plan to release our full code including training and evaluation as well as all model checkpoints for the camera-ready version of the paper.**

**LanguageBind** We use the code from the original author, using the pre-trained weight `LanguageBind/LanguageBind_Video_merge` uploaded on `huggingface`.

*Table 4.* **Encoder Information**. Detailed information about the four encoders used in our experiments. They represent a broad coverage of visual information and training objectives.

| Model | Architecture | Expertise | Training Datasets | Training Objective |
|---|---|---|---|---|
| LanguageBind (Zhu et al., 2024a) | ViT-L/14 | Video+Language | VIDAL-10M, five-modal video examples | Contrastive |
| DINOv2 (Oquab et al., 2023) | ViT-L/14 | Spatial | LVD-142M | Self-Supervised |
| ViViT (Arnab et al., 2021) | ViViT-B/16×2 | Actions/Temporal | Kinetics-400/600, short videos | Supervised |
| SigLIP (Zhai et al., 2023) | ViT-B/16 | Image+Language | 4B curated image/text pairs | Contrastive |

**DINOv2** As DINOv2 is an image-model, we get embedding per frame, and concatenate them to be a video embedding. We use $\mathrm{ViT_{Large}}$ model, pre-trained on LVD-142M dataset, and take the penultimate layer for the embeddings. Specifically, we use `timm`'s `vit_large_patch14_reg4_dinov2.lvd142m`

**ViViT** We use $\mathrm{ViT_{base}}$ as our backbone, pre-trained on Kinetics-400 dataset. Specifically, we use `google/vivit-b-16x2-kinetics400` uploaded on `huggingface`. We use featurizer output as the video embedding.

**SigLIP** As SigLIP is an image-model, we get embedding per frame, and concatenate them to be a video embedding. We use $\mathrm{ViT_{base}}$ as our backbone, and take the penultimate layer for the embeddings. Specifically, we use `timm`'s `vit_base_patch16_siglip_224`

We also considered multiple other options for encoders, such as CLIP-ViP (Xue et al., 2023) for our video-language contrastive expert, V-JEPA (Bardes et al., 2024) and Hiera (Ryali et al., 2023) for our pure video model, and CLIP (Radford et al., 2021) for our image-language contrastive expert, but found that our choices performed better overall.

### A.3. Detailed Experimental Results

Here we tabulate the full experimental results that was abbreviated from the main paper. The first table (Table 5) ablates the different training recipes we tried for MERV, with extended discussion in Section 4.2.3.

*Table 5.* **Ablation of training stage recipes**. We explore different training recipe strategies, starting with the standard LLaVA recipe which Video-LLaVA adopted, along with some other variations.

| Methods | MSVD-QA | | MSRVTT-QA | | TGIF-QA | | Perception Test | ActivityNet-QA | |
|---|---|---|---|---|---|---|---|---|---|
| | Acc | Score | Acc | Score | Acc | Score | Acc | Acc | Score |
| MERV (frozen) | **70.97** | 3.76 | **59.03** | **3.25** | 51.1 | 3.26 | 46.21 | **50.87** | **3.34** |
| MERV, Video-LLaVA recipe | 70.92 | 3.78 | 58.74 | 3.25 | 51.67 | 3.27 | 47.48 | 50.42 | 3.33 |
| MERV (full) | 70.48 | **3.79** | 57.25 | 3.24 | **51.39** | **3.28** | **48.41** | 49.93 | 3.33 |
| MERV, mixed Stage 1+2 | 69.9 | 3.73 | 55.14 | 3.08 | 51.53 | 3.26 | 45.65 | 39.98 | 2.95 |

**Pre-fusion projector details.** The image-level projectors are similar to those described in MM-1 (McKinzie et al., 2024): 2D adaptive average pooling, a shallow attention resampler similar to a Perceiver Resampler (Alayrac et al., 2022), and convolutional pooling with 3 RegNet blocks on both sides of an average pool layer such as the C-Abstractor in Honeybee (Cha et al., 2024). For video-level projectors, we use a 3D average pool, where we pool to the same spatial dimension but furthermore pool the frame dimension by 2, and a 3D convolution where we add a single $2 \times 3 \times 3$ convolution before the same average pooling. For all projectors, we project to the same number of tokens $t \times h \times w$, using an adaptive average pool or $h \times w$ latent tokens for the attention resampler.

We also provide the full metric numbers in Table 6a, 6b and 6c for our ablations that are described in Sections 4.2.1 and 4.2.2.

**Note on TGIF-QA** Video-ChatGPT's and Video-LLaVA's author-reported numbers on TGIF are incomparable as they were on a subset of the dataset. See `https://github.com/PKU-YuanGroup/Video-LLaVA/issues/37`.

*Table 6.* **Full design choice ablation numbers.** Detailed experimental results of Tables 2a, 2b, 2c. We highlight our defaults in orange and **bold** the best results.

(a) **Pre-fusion projectors.** * is 16 frames instead of 8. Top two rows are projector-free baselines.

| Projector | MSVD | MSRVTT | TGIF | Perc. | Params | FLOPs |
|---|---|---|---|---|---|---|
| 257 tok | 68.47 | 55.81 | 48.62 | 46.14 | - | - |
| class tok | 65.98 | 55 | 43.7 | 43.51 | - | - |
| 2D Avg | 68.23 | 56.92 | 48.99 | 45.69 | 0 | 2.1M |
| 2D Avg* | **69.08** | **58** | **50.01** | 46.34 | 0 | 4.2M |
| 2D Attn | 65.76 | 55.23 | 43.35 | 44.14 | 12.7M | 9.7G |
| 2D Conv | 67.48 | 56.78 | 47.6 | 45.04 | 237M | 241G |
| 3D Avg* | 68.62 | 57.2 | 49.59 | 44.95 | 0 | 4.2M |
| 3D Conv | 68.56 | 57.03 | 49.28 | **46.81** | 113M | 232G |

(b) **Pre-fusion output token.** We ablate the optimal token size per frame for the pre-fusion projector.

| Tkns | MSVD | MSRVTT | TGIF | Perc. |
|---|---|---|---|---|
| 1 | 61.94 | 54.64 | 41.41 | 42.85 |
| 4 | 64.47 | 55.72 | 45.32 | 43.31 |
| 16 | 67.23 | 56.44 | 47.75 | 43.18 |
| 64 | **69.08** | **58.00** | **50.01** | **46.34** |
| 100 | 68.38 | 57.47 | 48.78 | 45.56 |
| 144 | 68.65 | 57.73 | 48.81 | 43.94 |
| 256 | 68.46 | 57.72 | 48.66 | 43.51 |

(c) **Feature fusion strategy.** We compare our feature fusion strategy with concatenating the visual embeddings in either token sequence dimension or the channel dimension, learning an optimal embedding mixture weights, and training with equal 25% mixture of visual embeddings.

| Strategy | MSVD | MSRVTT | TGIF | Perc. | FLOPs |
|---|---|---|---|---|---|
| Cross-Attn | **70.97** | **59.03** | **51.1** | 46.21 | 17.19 T |
| Concat (Seq.) | 66.99 | 56.95 | 48.20 | 45.67 | 43.09 T |
| Concat (Ch.) | 70.02 | 58.08 | **51.1** | **47.36** | 16.29 T |
| Learnable W | 68.06 | 56.54 | 48.82 | 46.6 | 16.24 T |
| 25% - Mixed | 68.38 | 56.99 | 47.71 | 43.66 | 16.39 T |

*Table 7.* **Effect of Each Encoder**. Detailed results of Figure 4a.

| Methods | MSVD-QA | | MSRVTT-QA | | TGIF-QA | | Perception | ActivityNet-QA | | Avg |
|---|---|---|---|---|---|---|---|---|---|---|
| | Acc | Score | Acc | Score | Acc | Score | Acc | Acc | Score | Acc |
| All 4 encoders | **70.97** | **3.76** | **59.03** | **3.25** | **51.10** | **3.26** | 46.21 | 50.87 | **3.34** | **55.64** |
| w/o LanguageBind | 68.52 | 3.69 | 57.10 | 3.19 | 50.20 | 3.23 | 45.23 | 49.78 | 3.31 | 54.17 |
| w/o DINOv2 | 69.75 | 3.74 | 57.70 | 3.23 | 49.94 | 3.23 | 46.57 | **51.43** | **3.34** | 55.08 |
| w/o ViViT | 70.12 | 3.75 | 58.26 | 3.23 | 50.45 | 3.22 | **46.94** | 51.36 | 3.33 | 55.43 |
| w/o SigLIP | 69.85 | 3.74 | 57.55 | 3.22 | 50.27 | 3.22 | 46.20 | 50.06 | 3.32 | 54.79 |

*Table 8.* **MERV Captures Single Encoder Performances**. Detailed experimental results of Figure 4a.

| Methods | MSVD-QA | | MSRVTT-QA | | TGIF-QA | | Perception | ActivityNet-QA | | Avg | | Params |
|---|---|---|---|---|---|---|---|---|---|---|---|---|
| | Acc | Score | Acc | Score | Acc | Score | Acc | Acc | Score | Acc | FLOPs | Overall |
| MERV | **70.97** | **3.76** | **59.03** | **3.25** | **51.10** | **3.26** | **46.21** | 50.87 | **3.34** | **55.64** | 17.19 T | 7686.0 M |
| LangBind | 68.47 | 3.71 | 55.81 | 3.16 | 48.62 | 3.19 | 46.14 | 44.72 | 3.17 | 52.75 | 41.3 T | 7147.0 M |
| DINOv2 | 65.44 | 3.62 | 53.46 | 3.09 | 41.53 | 2.96 | 42.73 | 43.39 | 3.09 | 49.31 | 40.88 T | 7046.0 M |
| ViViT | 59.95 | 3.43 | 51.81 | 3.05 | 38.1 | 2.84 | 40.2 | 43.98 | 3.16 | 46.81 | 27.12 T | 6830.0 M |
| SigLIP | 66.68 | 3.64 | 56.41 | 3.16 | 48.22 | 3.16 | 45.09 | 49.41 | 3.31 | 53.16 | 31.08 T | 6834.0 M |

*Table 9.* **Performance on WH-words**. Detailed experimental results of Figure 6.

| | MSRVTT-what | MSRVTT-who | MSRVTT-how | MSRVTT-when | MSVD-what | MSVD-who | TGIF-what | TGIF-how | TGIF-where |
|---|---|---|---|---|---|---|---|---|---|
| MERV | **50.62** | **77.17** | 83.96 | 72.23 | **62.68** | **84.62** | **49.44** | **53.33** | **65.34** |
| ViViT | 43.06 | 70.43 | 78.90 | 68.54 | 50.10 | 75.90 | 32.90 | 50.10 | 50.62 |
| DINOv2 | 44.54 | 73.00 | 76.95 | 65.73 | 55.71 | 82.12 | 37.14 | 50.69 | 58.40 |
| LanguageBind | 46.89 | 74.86 | 83.41 | **72.53** | 59.66 | 82.95 | 46.23 | 53.24 | 59.92 |
| SigLIP | 47.96 | 74.41 | **84.21** | 71.20 | 57.17 | 82.23 | 45.65 | 53.25 | 60.21 |

## A.4. Something-Something v2 Details

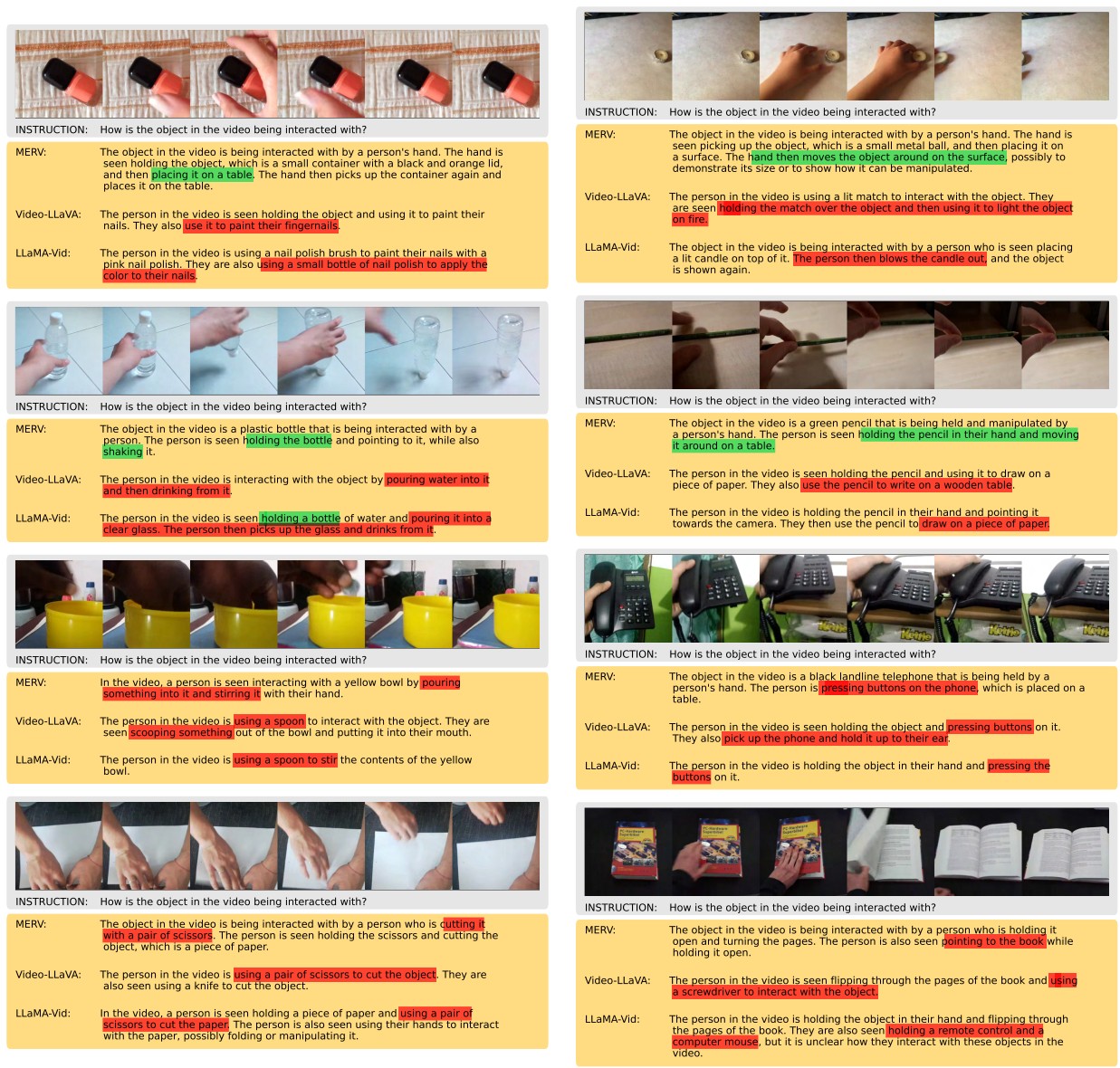

*Figure 8.* **Samples of MERV in SSv2.** Due to our design, our method shows better temporal action understanding than other VideoLLMs. (Top two rows) However, due to the difficulty of the task, we see failure cases for VideoLLMs. (Bottom two rows)

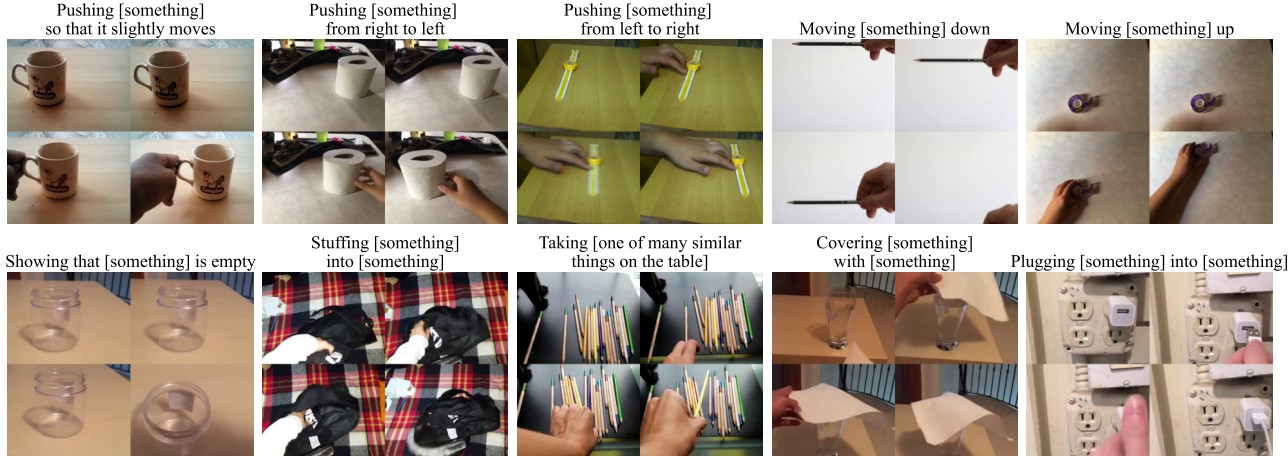

*Figure 9.* **Example video of Something-Something V2**. We see that ViViT show better performance in classes where temporal movement is critical for solving the task (Top row), while SigLIP performs better when the action can be inferred from the image without temporal information (Bottom row).

### A.4.1. SOMETHING-SOMETHING V2 - OPENENDED

*Table 10.* **Performance on Something-Something V2 - OpenEnded**. These are the performance in shown Figure 4b

|  | MERV | MERV-Full | LanguageBind | DinoV2 | ViViT | SigLIP | LLaMA-Vid-7B | LLaMA-Vid-13B | VideoLLaVA |
|---|---|---|---|---|---|---|---|---|---|
| Smth-Smth V2-OE-Temporal | 6.82 | **9.13** | 3.63 | 3.88 | 5.50 | 4.25 | 6.07 | 3.94 | 5.57 |
| Smth-Smth V2-OE | 17.70 | **20.65** | 13.83 | 11.03 | 10.53 | 13.84 | 16.47 | 15.62 | 19.18 |
| Smth-Smth V2-MCQ-Temporal | 36.84 | **40.65** | 30.58 | 28.08 | 39.77 | 25.39 | 27.14 | 17.89 | 22.47 |
| Smth-Smth V2-MCQ | **42.01** | 39.76 | 36.82 | 33.06 | 26.78 | 34.86 | 36.63 | 39.43 | 23.14 |

Additionally, we evaluate Something-Something V2 as an open ended QA task, where the question is "*How is the object in the video being interacted with?*", and the answer is expected to be similar to the class label. We use Video-ChatGPT (Maaz et al., 2024)'s LLM evaluation for validating the VideoLLMs' output. Table 10 tabulates the results. We see similar conclusions as with MCQ where ViViT excels at temporal subset than other single-encoder LLMs, but failing to match the performance in full dataset. Nonetheless, our method shows the best performance overall.

### A.4.2. SOMETHING-SOMETHING V2 - TEMPORAL

The 12 selected classes are as following:

- Approaching [something] with your camera
- Turning the camera downwards while filming [something]
- Turning the camera left while filming [something]
- Turning the camera right while filming [something]
- Turning the camera upwards while filming [something]
- Moving away from [something] with your camera
- Moving [something] away from the camera
- Moving [something] towards the camera
- Pulling [something] from left to right
- Pulling [something] from right to left
- Pushing [something] from left to right
- Pushing [something] from right to left

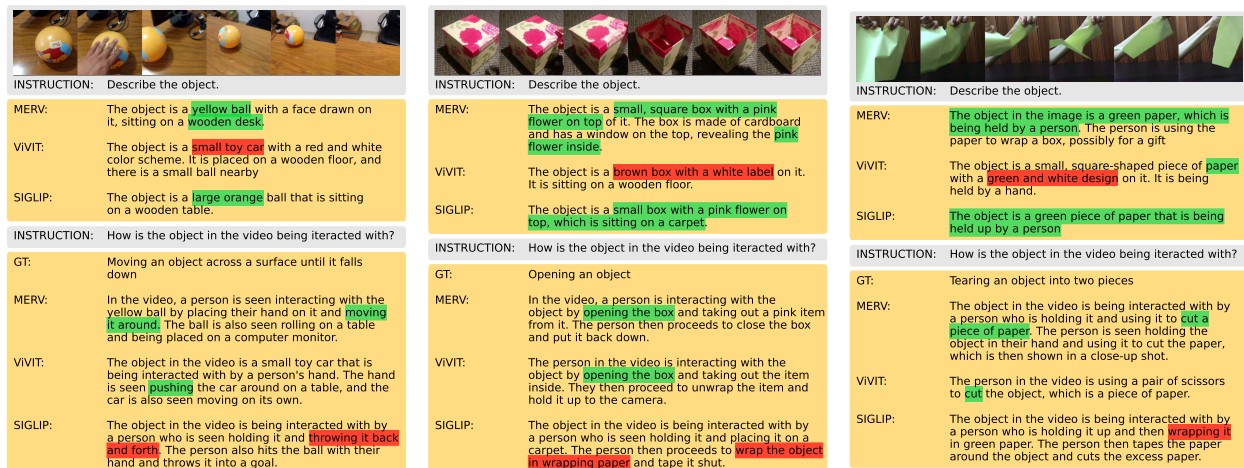

Figure 10. **Example VideoLLM output on Something-Something v2**. While SigLIP performs better on object and scene recognition, it fails to understand temporal actions. ViViT fails on the details of object recognition, but has better understanding in temporal movements.

## A.5. Additional Experiments and Analyses

### A.5.1. ADDING MORE ENCODERS

Additionally, we test with one more encoder Hiera (Ryali et al., 2023). The checkpoint we use was pretrained using the Masked Autoencoder (MAE) self-supervised learning technique on Kinetics 400 (K400), and then finetuned on K400 using supervised learning.

Table 11. **Performance with Hiera (Ryali et al., 2023)**.

| Methods | MSVD-QA | | MSRVTT-QA | | TGIF-QA | | Perception | ActivityNet-QA | |
| | Acc | Score | Acc | Score | Acc | Score | Acc | Acc | Score |
|---|---|---|---|---|---|---|---|---|---|
| VideoLLaVA | 67.74 | 3.69 | 56.90 | 3.18 | 47.99 | 3.17 | 47.08 | 3.27 | 44.22 |
| MERV | **70.97** | **3.76** | **59.03** | **3.25** | **51.1** | **3.26** | 50.87 | **3.34** | 46.21 |
| MERV, ViViT replaced with Hiera | 69.68 | 3.74 | 57.64 | 3.22 | 50.38 | 3.24 | 50.24 | **3.34** | **47.50** |
| MERV + Hiera | 69.67 | 3.72 | 58.26 | 3.23 | 50.32 | 3.22 | **51.23** | **3.34** | 46.23 |
| ViViT Single Encoder LLM | 59.95 | 3.43 | 51.81 | 3.05 | 38.10 | 2.84 | 43.98 | 3.16 | 40.20 |
| Hiera Single Encoder LLM | 55.38 | 3.28 | 49.21 | 2.95 | 36.02 | 2.76 | 44.01 | 3.15 | 40.20 |

The results are presented in the Table 11, with the best result in bold. Replacing ViViT with Hiera demonstrates improvements on the fine-grained spatio-temporal reasoning benchmark, Perception Test, with accuracy gain of 1.29%. Similarly, adding Hiera yields an improvement on ActivityNet, achieving a 0.36% increase in accuracy. However, on other benchmarks, the original MERV remains the strongest model. Overall, we observe no significant performance improvement when training with Hiera, which aligns with expectations, since Hiera is under the same paradigm as ViViT, functioning as a temporal expert trained on short videos. We also hypothesize that Hiera is more sensitive to the temporal stride than ViViT, as ViViT can reasonably deduce motion from uniformly sampled frames. We expect performance to improve if we incorporate encoders trained on different paradigms and data sources or process a much greater number of frames simultaneously, which we will leave for future work.

### A.5.2. ATTENTION WEIGHTS

Our analysis in Section 5.1 looks at how different cross-attention weights activate most on different videos, illustrated in Figure 11.

## A.6. More Qualitative Results

We present additional qualitative results in Figure 12 and Figure 13.

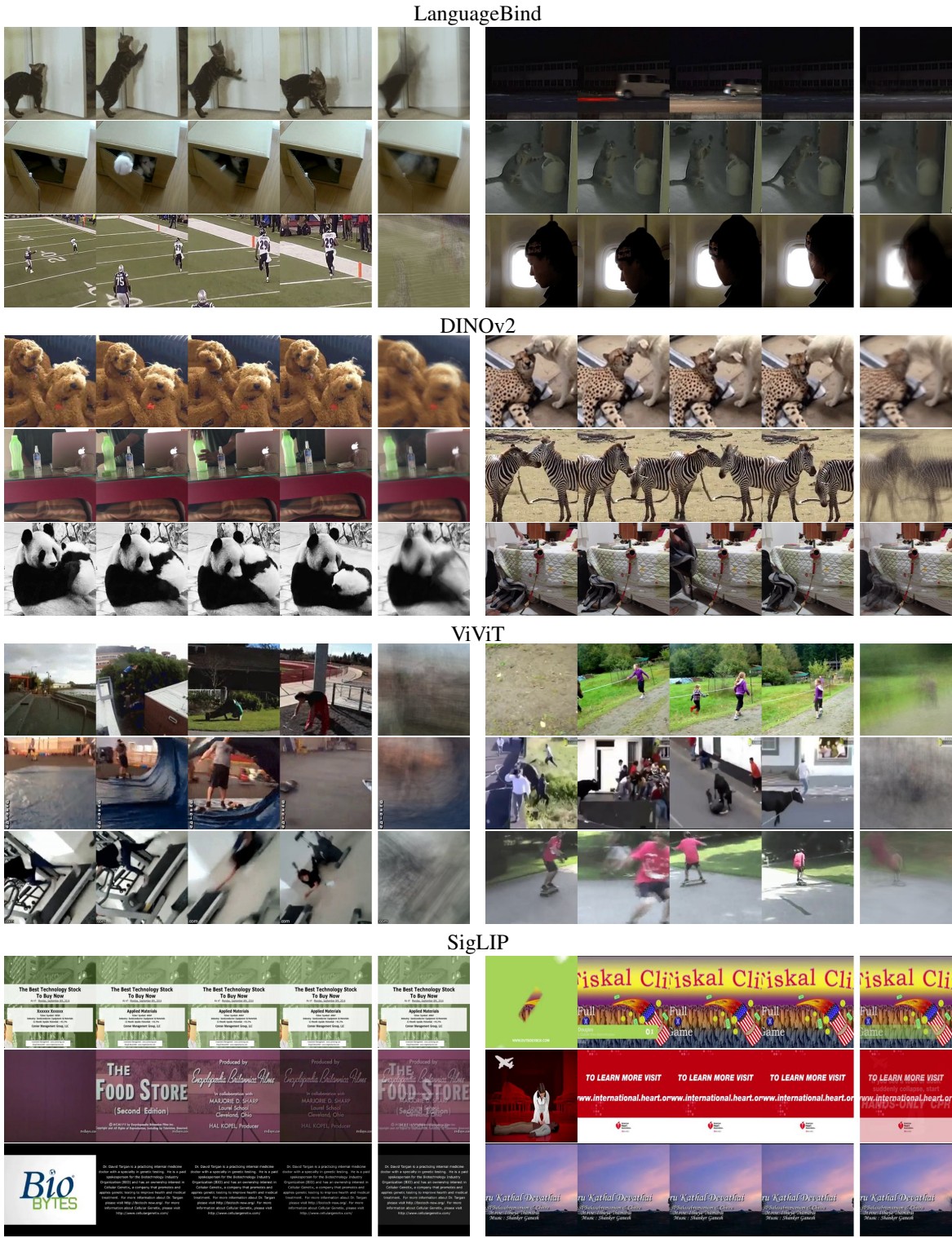

*Figure 11.* **Videos that give the highest attention weight for each of the encoders.** The right-most column shows the average frame of the video.

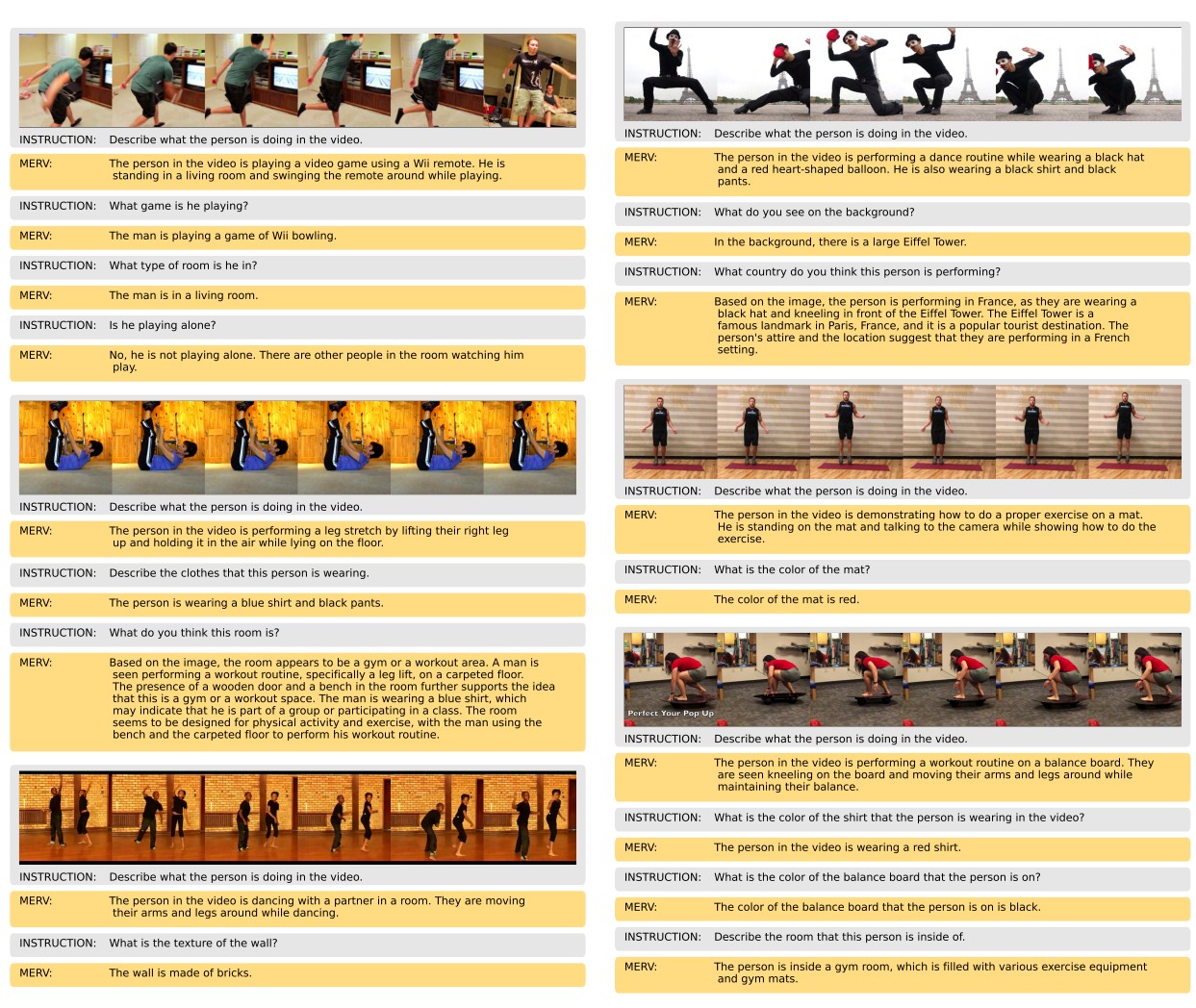

*Figure 12.* Samples of MERV in video understanding.

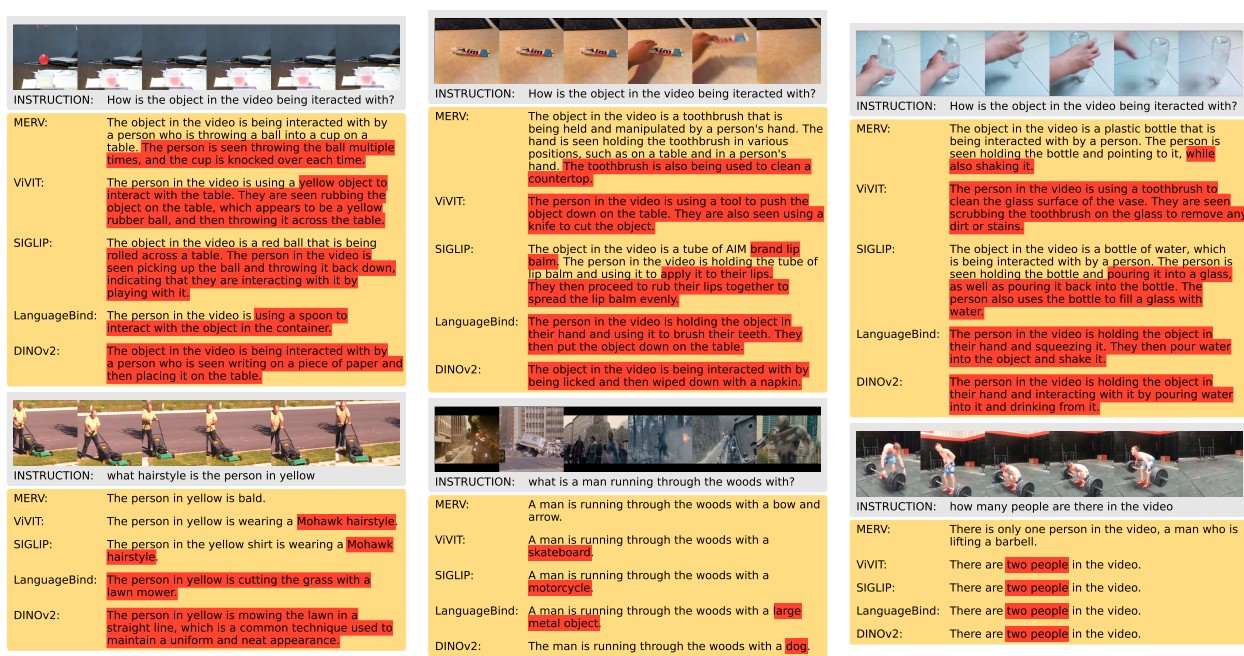

*Figure 13.* **More qualitative results**. MERV tends to show improved understanding in temporal-heavy videos as in Something-Something v2 dataset (Goyal et al., 2017) (Top Row), while retaining the performance on scenic understanding, seen from popular video benchmarks (Xu et al., 2017; Yu et al., 2019) (Bottom Row).

