# OpenReview forum: "Unifying Specialized Visual Encoders for Video Language Models"
_ICML.cc/2025/Conference — ICML 2025 poster_

### Official Review · Reviewer_jZRP · 2025-03-10

**Overall Recommendation:** 4

**Summary:**

The paper presents a VideoLLM that aligns and unifies the outputs of four different visual encoders to improve its video understanding capabilities. Specifically, the authors use a spatial encoder (DINOv2), a temporal encoder (ViViT), an image-language encoder (SigLIP), and a video-language encoder (LanguageBind). They design the inputs to the four encoders to be spatio-temporally aligned and train a pre-fusion projection network followed by a cross-attention network to respectively transform the features from the different encoders into a common space and then combine those features. The authors perform relevant experiments to evaluate their approach on multiple video Q&A datasets and compare against baseline methods.

**Claims And Evidence:**

The central claim -- that of combining different visual encoders focusing on different aspects of image, video, and their correlations with text inputs to improve video understanding performance -- is sufficiently backed up by experimental results. However, the additional benefits of computational efficiency can be evidenced further:

1. In Table 1, what are the total running times and FLOPs of the compared methods? How much does the proposed method improve on this aspect?

2. Since the computational improvements reported in Fig. 4a are 3-7 orders of magnitude higher than the FLOPs required for the pre-fusion projectors (Table 2a), can the accuracy numbers be improved further (while keeping the computational increases insignificant) by considering longer sequences as inputs to the pre-fusion projectors? Or would longer sequences also significantly impact the computational overheads of the feature fusion step (Table 2c)?

**Essential References Not Discussed:**

While not a domain expert, I did not find any major missing references in my search.

**Experimental Designs Or Analyses:**

The experiments, including the ablations, are sound, and the analyses are generally thorough. One observation from the results (Fig. 5 in particular) is that the proposed approach improves significantly over DINOv2 and ViViT across the board, but marginally over SigLIP and LanguageBind in many categories. To complement these results, looking at failure cases due to removing individual encoders (especially DINOv2 and ViViT) can further help understand their specific contributions.

**Methods And Evaluation Criteria:**

The proposed method is technically sound, and the evaluation criteria are appropriate.

**Other Comments Or Suggestions:**

I am unclear on the authors' example of an action that is indistinguishable from itself if temporally reversed. Pulling from left to right is temporally distinct from pulling from right to left as (a) the hand location relative to the object is different, and (b) the direction of movement is reversed (Fig. 8, columns 2 and 3). However, pulling from left to right, if temporally reversed, can be indistinguishable from *pushing* from right to left. Was that the intended meaning of "indistinguishable"? Maybe the authors can make this point fully clear with a pair of temporally indistinguishable actions in Fig. 8?

**Other Strengths And Weaknesses:**

N/A

**Questions For Authors:**

Please refer to the comments in previous sections.

**Relation To Broader Scientific Literature:**

With the development of language models for video understanding, the paper's contribution is relevant and timely, and it establishes new baselines for future video understanding models. It will likely interest the broader scientific communities working on language models, video understanding, and their intersections.

**Theoretical Claims:**

Not applicable - the paper presents experimental findings to justify the proposed approach.

---

> ### Author Rebuttal · Authors · 2025-04-01
>
> Dear Reviewer jZRP, we thank you for spending your time and effort on reviewing our work. We appreciate your recognition of our thorough experimental setup and ablations that sufficiently backs up our claim, as well as its relevant and timely contribution towards a broader collection of works regarding video understanding models. We will address your two main questions, as well as the two other comments you had for our work.
>
> **Q1: Total running time and FLOPs of compared methods?** \
> We obtained the running time and FLOPs for the methods in Table 1. In the main paper, we have calculated FLOPs by fixing the length of the textual tokens. But due to different generation schemes, this method cannot be applied to some VLMs.
>
> To make a fair comparison between VLMs, we use the same video and with the prompt "Describe this video." We follow the prompt builder and the video-processor provided by the official implementation. We report the Time-to-first-token (TTFT) as a metric for running time. Our FLOPs were measured in the same TTFT setup.
>
> |  | TTFT (Inference) | TFLOPs |
> |---------------|------------------|--------|
> | Video-Chat	| 135ms| 3.22   |
> | LLaMA-Adapter | 124ms| 2.69   |
> | Video-LLaMA   | 169ms| 11.07  |
> | Video-ChatGPT | 321ms| 10.69  |
> | SeViLA^  | X (two LMs)  	| X  	|
> | LLaMA-VID-7B  | 217ms | 3.05   |
> | LLaMA-VID-13B | 242ms| 3.47   |
> | Video-LLaVA   | 312ms | 15.93  |
> | MERV**  | 261ms | 10.5   |
>
> ^ Since SeViLA is MCQ-only VLM, needing to run LLM per each MCQ option, we do not compute TTFT and FLOPs as we cannot compare them with other models in a fair manner. \
> ** As we are using a different prompt, the FLOP value is different from the main paper.
>
> Our method, despite having 4 encoders, has far fewer FLOPs compared with Video-LLaVA (which our work primarily builds on), due to the compressed visual representation used. While LLaMA-VID achieves fewer FLOPs by restricting a single frame to have 2 visual tokens, we can achieve a much better performance with a similar step time with our efficient implementation.
>
> **Q2: Can accuracy be improved by using longer input sequences to the pre-fusion projector, and how would it affect the computational overhead of feature fusion?**
>
> This is a very interesting question! Below we tried using inputs with longer sequences to the pre-fusion projector by scaling up temporal resolution (as spatial resolution is fixed at 224x224) to see if our method improves.
> |	MERV   |  TFLOPs |  MSVD | MSVD-Score | MSRVTT | MSRVTT-Score |  TGIF | TGIF-Score | Perception |
> |:---------:|:------:|:-----:|:----------:|:------:|:------------:|:-----:|:----------:|:----------:|
> | 16 frames | 17.19 | **61.61** |	**3.41**	|  **45.17** | 	**2.75** 	| **46.11** |	**2.65**	|	46.21   |
> | 32 frames | 26.23 | 61.28 |	3.40	|  44.75 | 	2.74 	| 45.42 |	2.62	|	**47.77**   |
> * _As gpt-3.5-turbo-0613 is now deprecated, we use GPT-4o-mini for open-ended evaluation._
>
> We find that our method fails to take advantage of the extra temporal resolution for most datasets except for the Perception Test (which is a temporally challenging dataset). This indicates an upper limit to the capacity of our LLM for processing extra visual information, in which case maybe extra token selections or compression needs to happen.
>
> Regarding the second part of the question, increasing the input sequence length (i.e., larger $l$) to the pre-fusion projector leads to a linear increase in computational overhead for the feature fusion step, as can be seen from Eq. 2. However, for the subsequent LLM processing, the overhead grows quadratically with sequence length.
>
> **C1: Looking at failure modes when removing DINOv2 and ViViT**
>
> Thank you for the suggestion, we really appreciate it! We have not thought of the qualitative results of visualizing the failure cases when removing ViViT from MERV. We hypothesize that the failed cases are the videos that contain a lot of temporal movements, e.g., rapid camera movement. The hypothesis comes from a similar experiment, Figure 10 in Appendix, where we look at the attention weight of the feature fusion module, and see which video gains the highest weight for each encoder. As we saw that it was often the video that has large movements that is preferred by ViViT, we believe that these videos will have the most impact when ViViT is removed from MERV. We will add the suggested analyses in the camera-ready version of the paper.
>
> **C2: What is "temporally indistinguishable classes" in Something-Something V2 dataset?**
>
> Yes, your interpretation is exactly right. In particular, we selected action classes where the temporal reverse of one class approximates another class — as you pointed out, reversing frames of "Pulling from Left to Right" approximates frames of "Pushing from Right to Left". We identified 12 such class pairs from SSv2, which are listed in the appendix. We understand that our original explanation could be clearer, and we'll revise the paper following your helpful suggestion!

---

### Official Review · Reviewer_44Cd · 2025-03-11

**Overall Recommendation:** 5

**Summary:**

The authors propose to use multiple video encoders (instead of a single encoder) for visual feature extraction in the context of video-LLMs. They propose a simple, straightforward technique to ensemble multiple encoders with a clever feature fusion strategy leading to lesser inference-time FLOPs than any of the individual encoders. Thorough and extensive evaluations establish the usefulness of proposed method while also uncovering some interesting insights on visual encoders for video tasks.

**Claims And Evidence:**

Yes. Extensive experimentation demonstrates usefulness of proposed method.

**Essential References Not Discussed:**

N/A

**Experimental Designs Or Analyses:**

Authors follow standard experimental settings on established benchmarks.

**Methods And Evaluation Criteria:**

Yes. Clear method with extensive evaluation.

**Other Comments Or Suggestions:**

1. On L125 (right column): "As t is the same across each v_e" - why is this? do the video encoders used have no pooling / compression across time axis?
2.  Eq (2) for feature fusion: can you discuss motivation for this exact design choice?
3. Table 2 (b) - could these numbers be reported (maybe in appendix) for Perception test and SSv2?
    * Could more tokens help with more temporally complex tasks?
    * This is motivated by ViViT outperforming MERV on several such tasks. Could the lower token count (information bottleneck) of MERV be a reason for that?
4. The SSv2 analysis against ViViT, maybe repeat with same token count for MERV? It maybe unfair for MERV to compete against a more expensive ViViT version that uses more TFLOPs.
5. What is the black outer-most line in Figure 1 (right)? This is unclear.

**Other Strengths And Weaknesses:**

**Strengths**
1. The paper is well written with clear presentation of methodology and experiments.
2. The authors explore a timely topic (visual encoders for video) that could be valuable to the video understanding community.
3. A simple and straightforward modification to existing baselines with clever design choices to maintain inference compute.
4. Extensive experimentation achieving strong results improvements across a range of video QnA benchmarks.


**Weaknesses**
1. Minor improvements for better clarity (see comments / suggestions).

**Questions For Authors:**

1. Could the feature aggregation be conditioned on the textual prompt? Is this a direction that was explored at any point?
    * E.g. In feature fusion cross-attention operation, the queries could be derived / conditioned on from textual prompt.

**Relation To Broader Scientific Literature:**

Visual encoders are an important component in multi-modal LLMs (especially) video, and is relatively under explored. The authors investigate a novel direction of ensembling multiple visual encoders with negligible increase in runtime costs. This direction can be valuable to the video understanding community, especially for video QnA.

**Theoretical Claims:**

N/A

---

> ### Author Rebuttal · Authors · 2025-04-01
>
> Dear Reviewer 44Cd, thank you for your time and effort on reviewing our work. We are grateful for the insights that you have provided us. Specifically, we appreciate that you have seen our work to have **novel and clever strategy** with **less inference-time and FLOPs**, providing **interesting insights** on visual encoders for the underexplored field of video understanding. We appreciate that you see the main weakness of the paper to be in clarification, which we will address in order below along with your question.
>
> **C1: On L125 (Right), why is $t$ the same across each $v_e$?** \
> This is because, prior to spatial alignment, we performed temporal alignment. You are right that video encoders may have temporal pooling, which alters the temporal resolution. Therefore, we achieve temporal alignment in the most reliable and straightforward way—by adjusting the number of input frames fed into each visual encoder. E.g., we feed 32 frames to ViViT so that the output $t=16$.
>
> **C2: Can you discuss motivation for your feature fusion design?** \
> We wanted a design which could adaptively and efficiently select the most relevant visual information from all of the encoders. The simplest methods, such as sequence and channel concat, allow us to naively combine the information at its full resolution, but are inefficient and limited, e.g., sequence concat ignores our spatio-temporal alignment and orders in sequence. We can make this more efficient by summing the features together with learnable scalar weights for each, i.e., the “Learnable W” row in Table 2(c). This reduces the output sequence length. However, this has much weaker expressiveness than other methods and is still not adaptive.
>
> Cross-attention achieves the best of both worlds, being expressive while still being efficient. Using learnable queries as input, we adaptively set our encoder feature weights based on the input video and sum the features. The final output is of shape [BxLxD] instead of [Bx(NL)xD] where N is the number of encoders. Our ablations in Table 2(c) compare all of these feature fusion methods and find ours to be the most performant, along with channel concat. In fact, both methods are similar at heart. Channel concat creates a feature of shape [BxLx(ND)], but immediately has a linear layer following it to project into the LLM dimension of D. Our cross attention does a similar feature consolidation but constrained at the encoder level, i.e., each encoder’s feature gets one global weight.
>
> **C3: Can more output tokens from pre-fusion projector help more temporally complex tasks? Can the lower token count bottleneck MERV on Perception test and SSv2?**
>
> | | LanguageBind-LLM | MERV w/ 4 encoders |
> |:-:|:-----:|:-------:|
> |**Tkns**| Perception / SSv2 | Perception / SSv2 |
> |1|42.85/32.97|44.52/33.89|
> |4|43.31/33.34|43.16/35.81|
> |16|43.18/36.29|46.45/41.47|
> |64|44.43/37.36|46.21/**42.01**|
> |100|**45.56**/37.72|44.96/40.58|
> |144|43.94/**38.82**|45.86/39.40|
> |256|43.51/35.25|**47.26**/40.06|
>
> We evaluated LanguageBind-LLM and MERV on PerceptionTest and SSv2 across varying pre-fusion output token counts. The results indicate that while more tokens can help with more temporally complex tasks, performance gains generally plateau beyond 64 tokens, suggesting that 64 tokens offer an optimal trade-off between performance and efficiency. We agree that limited token count could lead to an information bottleneck depending on the complexity of the task at hand, but in our studies, 64 tokens seems to be enough.
>
> **C4: In Figure 4, it may be unfair for MERV to compare with ViViT-LLM where ViViT-LLM uses more TFLOPs.**
>
> To make the TFLOPs comparable, we perform an additional comparison where we use pre-fusion projector (pfp) with ViViT-LLM, so that the visual token size is now 64 per frame, instead of 196.
>
> |Model|TFLOPs|SSv2|SSv2-Temporal|
> |:---:|:----:|:--:|:-----------:|
> |MERV|17.19|**42.01**|36.84|
> |MERV (full)|17.19|39.76|**40.65**|
> |ViViT-LLM|27.12|26.78|39.77|
> |ViViT-pfp-LLM|13.00|29.07|35.15|
>
> As expected from Table 2(a), our pre-fusion projector is designed so that it does not hurt the performance while using fewer visual tokens. Regardless, our method shows better or matching performance with single-encoder ViViT, with or without pre-fusion projector.
>
> **C5: What is the black outermost line of Figure 1 (Right)?** \
> The thin black line is the edge of the graph. Thank you for spotting them. We will fix the graph to be more clear.
>
> **Q1: Can we make the queries text-conditioned? Did we try this?** \
> Yes, we tried. As of now, current Video-LLaVA training datasets have vastly different text distribution with the benchmarks. As we are doing zero-shot evaluation, we did try and find that our feature fusion module does not generalize to such out-of-distribution textual prompts when the queries fed to the fusion module. We think this could be an exciting scope for future research to effectively generalize a text-conditioning which works on broader examples.

---

> > ### Comment · Reviewer_44Cd · 2025-04-05
> >
> > Thanks to the authors for the comprehensive response. All clarification concerns have been resolved.
> >
> > The paper would definitely be valuable to the video understanding community, given its novel insights on building a *more performant ensemble* of multiple visual encoders, that is *computationally cheaper* for inference than any of the individual encoders, due to their clever fusion strategy.
> >
> > In light of above, I retain my rating of Strong Accept.

---

### Official Review · Reviewer_1K5Z · 2025-03-12

**Overall Recommendation:** 2

**Summary:**

Traditional VideoLLMs typically rely on a single vision encoder, which restricts the model's ability to leverage the diverse strengths of different visual encoders. To overcome this limitation, this paper propose a novel framework that integrates multiple specialized vision encoders into a unified video representation model. The proposed approach employs encoders such as SigLIP, DinoV2, ViViT, and LanguageBind—each contributing unique capabilities in spatial, temporal, and multimodal understanding.

**Claims And Evidence:**

Yes

**Essential References Not Discussed:**

N/A

**Experimental Designs Or Analyses:**

The authors conducted extensive experiments on combining visual encoders, covering both individual and joint usage scenarios. This exploration provides practitioners with insights into the effectiveness of different encoder combinations. Additionally, the authors analyzed the impact of various training stages in LLaVa-style video language model (VLM) training and offered observational conclusions on the importance of each stage in the two-stage training process.

**Methods And Evaluation Criteria:**

Strengths:

- The proposed method is very well motivated. Reproducibility is high.


Weaknesses:

- Although there is a performance improvement, my primary concern is that the innovation of the paper is rather limited. The paper proposes an empirical method for combining multiple visual encoders in a multimodal model. The feature fusion operation is based on cross-attention and linear projection layers. Since ensemble methods are a fundamental concept in machine learning, the paper does not provide new insights, and considering the increased computational cost, the performance improvement is expected and not surprising.

**Other Comments Or Suggestions:**

N/A

**Other Strengths And Weaknesses:**

N/A

**Questions For Authors:**

Q1: What are the reasons behind the authors' selection of these four encoders? For example, the chosen video model ViViT was released in 2021. However, many more advanced methods, such as MViTv2 and VideoMAE v2, are now available.

**Relation To Broader Scientific Literature:**

The authors enhanced the model's performance by integrating multiple visual encoders, an approach that is closely related to ensemble learning in machine learning.

**Theoretical Claims:**

The paper does not present any theoretical contributions. Instead, it employs an ensemble of multiple vision encoders to address the diverse tasks encountered in MLLMs.

---

> ### Author Rebuttal · Authors · 2025-04-01
>
> Dear Reviewer 1K5Z, we thank you for spending your time and effort on reviewing our work. We appreciate your recognition of the motivation, the novelty of our framework, extensive experiments, insights we provided to practitioners, and high reproducibility of our work. Your concerns seem to be with a) the **innovation** of our paper, particularly concerning our feature fusion and our relationship to ensemble methods, and b) our **reasoning behind our encoder selection** and the choice of picking more recent video encoders. Before we will address these in order, we want to point out an error in your review. **The supplementary material was provided in page 13~22**  which addresses some of your questions.
>
> **W1: Weak innovation about method, esp. around feature fusion and ensemble methods.**
>
> We agree that ensembling is a fundamental concept in machine learning, and simply applying such methods without understanding how each unique domain affects the method in practice would not provide any new insights. While our method may seem simple at a first glance, there were many design decisions which were critical in enabling the strength of our results which are not apparent if one tries to simply scale prior works to multiple encoders; see our detailed response to ukCZ, where we discuss how our method provides new insights into practically building these ensemble methods. In contrast, prior works such as Prismer [1]—though not focused on RGB-based video understanding MLLMs—have failed to make a marked improvement over single-encoder baselines.
>
> As for the feature fusion, our method is quite different from existing approaches, as those primarily fall under either channel concatenation or sequence concatenation, as mentioned in the paper text with references (Appendix A.2. Related Works). However, our method is still better or on par with these methods, while also providing extra, meaningful analyses into how the model is making these decisions as shown in Fig.10. By visualizing the videos that have the high attention weight of an encoder, e.g. videos with a lot of text have high attention weight on SigLIP, we gain insight of the data mix and training objective of each encoder.
>
> **Q1: Why these four encoders, and more recent video encoders?**
>
> We chose these four encoders because we felt that they offered a broad coverage of both optimization objectives and training data, both of which are important for giving our model a wide breadth of coverage. Within each type of model, we performed experiments using a few different models, such as V-JEPA [2] for video, but ended up choosing ViViT based on final performance. We hypothesize that V-JEPAl is difficult to use for frozen adaptation, so we leave this as future work for others. The other models were chosen based on strong performance in other places.
>
> In our Appendix A.6, we have conducted experiments using a newer SoTA video model Hiera-B+ [3], i.e. MViTv3, in both a four-encoder setting replacing ViViT and a five-encoder setting.
>
> For your convenience, we list the results in the table below, with the best result in bold and the second-best in italics. Replacing ViViT with Hiera demonstrates improvements on the fine-grained spatiotemporal reasoning benchmark, PerceptionTest, with accuracy gain of 1.29%. Similarly, adding Hiera yields an improvement on ActivityNet, achieving a 0.36% increase in accuracy. However, on other benchmarks, the original MERV remains the strongest model. Overall, we observe no significant performance improvement when training with Hiera, which aligns with expectations, since Hiera is under the same paradigm as ViViT, functioning as a temporal expert trained on short videos. We also hypothesize that Hiera is more sensitive to the temporal stride than ViViT, as ViViT can reasonably deduce motion from uniformly sampled frames. We expect performance to improve if we incorporate encoders trained on different paradigms and data sources or process a much greater number of frames simultaneously, which we will leave for future work.
>
> | Method | MSVD -Acc | MSVD -Score | MSRVTT -Acc | MSRVTT -Score | TGIF -Acc | TGIF -Score | ActivityNet -Acc | ActivityNet -Score | Perception -Acc |
>  | --- | --- | --- | --- | --- | --- | --- | --- | --- | --- |
>   | VideoLLaVA | 67.74 | 3.69 | 56.90 | 3.18 | 47.99 | 3.17 | 47.08 | 3.27 | 44.22 |
>  | MERV | **70.97** | **3.76** | **59.03** | **3.25** | **51.1** | **3.26** |  *50.87* | **3.34** | 46.21 |
> | MERV,  ViViT replaced with  Hiera | *69.68* | *3.74* | 57.64 | 3.22 | *50.38* | *3.24* | 50.24 | **3.34** | **47.50** |
>  | MERV + Hiera | 69.67 | 3.72 | *58.26* | *3.23* | 50.32 | 3.22 | **51.23** | **3.34** | *46.23* |
>
> [1] Liu et al., Prismer: A Vision-Language Model with Multi-Task Experts, 2023. \
> [2] Bardes et al. "Revisiting feature prediction for learning visual representations from video." arXiv 2024. \
> [3] Ryali et. al., Hiera: A Hierarchical Vision Transformer without the Bells-and-Whistles, ICML 2023.

---

### Official Review · Reviewer_ukCZ · 2025-03-14

**Overall Recommendation:** 2

**Summary:**

This paper tackles the problem of video-language understanding by introducing a multi-encoder strategy for constructing a comprehensive representation for videos. The authors claim that existing single-encoder methods can merely obtain a limited amount and type of visual information. Therefore, they proposed to leverage multiple encoders from different backbone families (assumed to have different and diverse capabilities) and map their representations into a unified space. Experimental results demonstrate the performance gains given by such multi-encoder design.

**Claims And Evidence:**

The main claim in the paper is “existing single-encoder methods are capable of obtaining only a limited amount and type of visual information.”, which is somewhat reasonable, but it is challenging to proof it with any rigorous theoretical analysis.

**Essential References Not Discussed:**

No

**Experimental Designs Or Analyses:**

The experiments are conducted on public video understanding benchmarks, with well-designed metrics and fair comparisons. The experiment protocols are reasonable.

**Methods And Evaluation Criteria:**

Yes. From the application perspective, the proposed strategy can effectively enhance the capabilities of existing MLLMs.

**Other Comments Or Suggestions:**

N/A

**Other Strengths And Weaknesses:**

Generally, the paper provides some practical design choices and experiments on how to extend single-encoder models to multi-encoder styles. Since this is a simple strategy that most researchers would know of its effectiveness, I think the novelty of this paper is limited. It would be better to provide more insights about how to jointly encode more diverse information (using a single encoder or only a few smaller encoders) rather than simply making use of more encoders.

**Questions For Authors:**

The authors are encouraged to provide more strong justifications for the novelty of the proposed framework.

**Relation To Broader Scientific Literature:**

The key contribution is to extend the existing Video-LLMs from single-encoder to multi-encoder, which I think is already a common strategy in enhancing visual perception capabilities in the entire CV community. This makes the novelty of the proposed scheme limited.

**Theoretical Claims:**

The paper does not contain any proofs or theoretical claims.

---

> ### Author Rebuttal · Authors · 2025-04-01
>
> Dear Reviewer ukCZ, we thank you for spending your time and effort on reviewing our work. We appreciate your positive comments on our practical design choices that can enhance the capabilities of existing Multimodal LLMs (MLLMs). Your concerns are with a) the **insights** gained from our paper, and b) the **novelty** of our method. We will address them as follows.
>
> We posit that while our final method seems simple, it is the result of exploring >100 different configurations, not all of which worked well, which resulted in this simplicity. For example, while prior methods like VideoLLaVA (which our work primarily builds on) used full-resolution visual tokens, we found that combining 2D averaging with just 64 tokens/frame outperformed this base setting and was far more efficient (Table 2(a) and (b)). Note that in Table 2(a), only 2D/3D Avg and 3D Conv outperform this base setting, with 3D Conv doing so at a significant cost in parameters and FLOPs. Similarly, in Table 2(b), 64 frame tokens outperform the default 256 from prior works, with 144 being the only other better setting. Finally, also see our response to Reviewer 1K5Z discussing how even our choice of encoders is not simple. This shows that it is not a matter of simply adding encoders, as without caring for these details, the resulting method is easily worse and even less efficient than the original method.
>
> As for novelty, our work is indeed the first to successfully leverage multi-encoders for video understanding MLLMs, as agreed by the Reviewer 1K5Z and 44Cd to be novel. We also believe there is merit in the detail and thought behind our paper. For example, we contribute a computationally efficient scheme for adding additional encoders which takes advantage of the distributed setups required today (Fig.3). We provide a very in-depth analysis of multi-encoder MLLMs, to a level of detail not taken by other works (Sec.3 and 4). We illustrate behaviors behind these large pretrained models reflecting their training data and inherent biases which are not commonly understood (Appendix, Fig.10) (e.g. SigLIP trained models are biased towards textual data). Additionally, one underrated choice we made was to deliberately focus on a single training mix so that we can control for data and understand these models and our choices. The video domain has many quirks not present in images, so discovering additional settings which work is nontrivial.
>
> Finally, we summarize our findings and share insights not covered in prior research:
>
> Insight 1: What works for integrating multiple RGB-encoders into a single VideoLLM:
> 1. Select encoders from distinct backbone families.
> 2. Align features spatio-temporally before fusion.
> 3. 2D average pooling (no parameters) is the best pre-fusion projector.
> 4. Optimal token size per frame is 64 (searched from 1 to 256).
> 5. Stage 2-only tuning is fast (43% time) with similar results; unfreezing LLM in Stage 1 boosts alignment and benchmark scores.
>
> Insight 2: What doesn’t work:
> 1. Assembling random encoders can hurt performance.
> 2. No pre-fusion strategies lead to worse results and higher FLOPs.
> 3. Other projectors (class token, 3D avg, perceiver-style attention, 2D/3D conv) perform worse and cost more.
> 4. Too small/large token sizes per frame degrade performance.
> 5. Alternative fusion methods (concat, equal weighting, learnable scalars) are suboptimal considering both performance and efficiency.
> 6. Training only projectors and fusion in Stage 1 or mixing stages performs worse than explicit two-stage training.
>
> Insight 3: Computational time doesn’t scale linearly with respect to the number of visual encoders used, as one might initially think, thanks to multi-GPU model parallelism. Using more encoders adds very minimal overhead (Fig.3) since they can run in parallel and are small compared to the LLM.
>
> Insight 4: MERV outperforms prior works by up to 4.62% with minimal added parameters and faster training. It leverages multiple encoders effectively, while each encoder is truly contributing, without trading off performance between specializations as single encoder models do. This offers a free alternative to scaling training data and model understanding by scaling visual encoders instead.
>
> Insight 5: MERV provides better interpretability with the inner workings of visual encoders and the decision-making process of MLLMs. Its cross-attention weights activate on corresponding videos, allowing it to capture the visual strengths of different encoders and intuit both motion and general understanding simultaneously.
>
> Summary: As elaborated above, we are not simply making use of more encoders, nor did MERV just provide a simple strategy that most researchers would already know to be effective. As Reviewer 44Cd and 1K5Z agrees, MERV introduces novel direction and is the result of extensive effort and care in exploring the right design choices. We provided many insights into how to effectively encode information using a single/few encoder(s).

---

### Decision · Program_Chairs · 2025-05-01

**Decision:**

Accept (poster)

**Comment:**

The manuscript received ratings of 5, 2, 4, and 2. Reviewers appreciated that the proposed method is well motivated and effectively enhances the capabilities of existing MLLMs following reasonable experimental protocols. Reviewers also raised questions including, limited technical novelty and the increased computational cost. Authors provided rebuttal to address reviewers concerns, especially improvements using the proposed approach without significant increase in the inference-time and FLOPs. Post-rebuttal, two reviewers remained positive whereas the other two persisted with their respective concerns. Given the reviews and rebuttal, the AC agrees with reviewer 44Cd and jZRP that the proposed approach is simple and is sufficiently backed up by experimental results in terms of performance gains without any significant increase in runtime costs. The simplicity and effectiveness of the proposed approach is also acknowledged by the other two reviewers (1K5Z and ukCZ). The reviewer 1K5Z further mentions that the reproducibility of the proposed aapproach is high and that the experimental exploration provides practitioners with insights into the effectiveness of different encoder combinations. The AC agrees with reviewers comments that the proposed study will be interesting for practitionars as it provides insights on building a more performant ensemble of multiple visual encoders which is computationally inexpensive at inference  due to the introduced fusion scheme, thereby being valuable to the video understanding community.